# Timing the initiation of multiple myeloma

Even H. Rustad [1,18], Venkata Yellapantula[1,18], Daniel Leongamornlert[2], Niccolò Bolli[3,4], Guy Ledergor [5], Ferran Nadeu [6,7], Nicos Angelopoulos [2], Kevin J. Dawson[2], Thomas J. Mitchell [2], Robert J. Osborne[2], Bachisio Ziccheddu[3,8], Cristiana Carniti[4], Vittorio Montefusco[4], Paolo Corradini [3,4], Kenneth C. Anderson[9], Philippe Moreau[10], Elli Papaemmanuil[11], Ludmil B. Alexandrov [12], Xose S. Puente [13,14], Elias Campo [6,7,13], Reiner Siebert[15], Herve Avet-Loiseau[16], Ola Landgren [1], Nikhil Munshi[9,17], Peter J. Campbell [2] & Francesco Maura [1,2✉]

The evolution and progression of multiple myeloma and its precursors over time is poorly understood. Here, we investigate the landscape and timing of mutational processes shaping multiple myeloma evolution in a large cohort of 89 whole genomes and 973 exomes. We identify eight processes, including a mutational signature caused by exposure to melphalan. Reconstructing the chronological activity of each mutational signature, we estimate that the initial transformation of a germinal center B-cell usually occurred during the first 2nd-3rd decades of life. We define four main patterns of activation-induced deaminase (AID) and apolipoprotein B mRNA editing catalytic polypeptide-like (APOBEC) mutagenesis over time, including a subset of patients with evidence of prolonged AID activity during the pre-malignant phase, indicating antigen-responsiveness and germinal center reentry. Our findings provide a framework to study the etiology of multiple myeloma and explore strategies for prevention and early detection.

[1] Myeloma Service, Department of Medicine, Memorial Sloan Kettering Cancer Center, New York, NY, USA. [2] The Cancer, Ageing and Somatic Mutation Programme, Wellcome Sanger Institute, Hinxton, Cambridgeshire CB10 1SA, UK. [3] Department of Oncology and Hemato-Oncology, University of Milan, Milan, Italy. [4] Department of Medical Oncology and Hematology, Fondazione IRCCS Istituto Nazionale dei Tumori, Milan, Italy. [5] Division of Hematology/Oncology, Department of Medicine, University of California, San Francisco, CA, USA. [6] Patologia Molecular de Neoplàsies Limfoides, Institut d'Investigacions Biomèdiques August Pi i Sunyer (IDIBAPS), 08036 Barcelona, Spain. [7] Centro de Investigación Biomédica en Red de Cáncer (CIBERONC), 28029 Madrid, Spain. [8] Department of Molecular Biotechnologies and Health Sciences, University of Turin, Turin, Italy. [9] Jerome Lipper Multiple Myeloma Center, Dana–Farber Cancer Institute, Harvard Medical School, Boston, MA, USA. [10] CRCINA, SIRIC ILIAD, University Hospital of Nantes, Nantes, France. [11] Computational Oncology Service, Department of Epidemiology & Biostatistics, Center for Computational Oncology, Memorial Sloan Kettering Cancer Center, New York, NY 10065, USA. [12] Department of Cellular and Molecular Medicine and Department of Bioengineering and Moores Cancer Center, University of California, La Jolla, San Diego, CA, USA. [13] Unitat Hematopatologia, Hospital Clínic of Barcelona, Universitat de Barcelona, 08036 Barcelona, Spain. [14] Departamento de Bioquimica y Biologia Molecular, Instituto Universitario de Oncologia (IUOPA), Universidad de Oviedo, Oviedo, Spain. [15] Institute of Human Genetics, Ulm University and Ulm University Medical Center, Ulm, Germany. [16] IUC-Oncopole, and CRCT INSERM U1037, 31100 Toulouse, France. [17] Veterans Administration Boston Healthcare System, West Roxbury, MA, USA. [18] These authors contributed equally: Even H. Rustad, Venkata Yellapantula. ✉email: mauraf@mskcc.org

Cancers are usually preceded by asymptomatic clonal entities that may be detected several years before progression to overt malignancy[1,2]. This suggests a long evolutionary process where genomic driver events accumulate over time, conferring advantage to distinct subclones, allowing their expansion and progression[3]. Understanding the timeline of premalignant clonal initiation and progression is vital to develop strategies for early cancer diagnosis and prevention. Intriguingly, recent reports have shown how different cancer types acquire the first driver event approximately 20–40 years before diagnosis, often when the individual is aged between 20 and 30 years[2,4,5]. Such studies have been made possible by the existence of mutational processes whose activity is constant over time (i.e., clock-like), producing a mutational burden proportional to the cancer cell age[4,6–9].

Multiple myeloma (MM) is always preceded by an asymptomatic expansion of clonal plasma cells, clinically recognized as monoclonal gammopathy of undetermined significance (MGUS) or smoldering myeloma (SMM)[10,11,12]. Historically, translocations between the *IGH* locus and recurrent oncogenes and trisomies of odd chromosomes (i.e., hyperdiploidy) have been considered as initiating events[13,14]. Hyperdiploidy is detectable in >60% of MM patients and has been thought to result from a single catastrophic mitosis[13]. In a recent whole-genome sequencing (WGS) study, we provided evidence that hyperdiploid cytogenetic profiles often reflect the sum of multiple gains acquired in different time windows[15]. This observation was based on the corrected ratio of duplicated and non-duplicated clonal point mutations within large chromosomal gains (i.e., molecular time)[2,15,16]. While this allowed us to re-construct the relative order in which events occurred, translating these estimates into an absolute time scale was not possible, because, in contrast to some solid cancers[5], MM does not show a reliable relationship between global mutation burden and patient age[6,17].

Here we characterize the landscape and temporal activity of mutational processes involved in MM pathogenesis, to reconstruct the evolutionary history in absolute time, and finally estimate the patient age at disease initiation.

## Results

**The MM mutational signature landscape.** To comprehensively assess the catalog of mutational processes involved in MM pathogenesis, we interrogated WGS data from 52 patients (Supplementary Data 1). Twenty-six patients (50%) had >1 sample collected at different time points for a total of 89 tumor samples (Supplementary Table 1). To improve the accuracy of de novo mutational signatures extraction, we integrated two independent approaches: SigProfiler and the hierarchical Dirichlet process (hdp; "Methods")[9,18]. Eight single-base substitution (SBS) signatures were identified, seven of which were compatible with one included in the most recent mutational signature catalog: SBS1, SBS2, SBS5, SBS8, SBS9, SBS13, and SBS18 (Fig. 1a; https://cancer.sanger.ac.uk/cosmic/signatures/SBS/)[9]. The final mutational signature, here named SBS-MM1, did not correspond to any of the COSMIC reference mutational signatures but was compatible with a recently reported MM mutational signature of unknown etiology (Fig. 1a and Supplementary Fig. 1A, B)[9,17–19]. The existence of SBS-MM1 as a distinct mutational process was further validated by extending the mutational signature analysis to include the two flanking bases 5' and 3' to the mutated base (1536 classes or 5-nucleotide context; Supplementary Fig. 1C, D).

Despite their statistical robustness, in heterogeneous cancer types such as MM, de novo extraction algorithms can be biased by the bleeding of mutational signatures between samples, where mutational signatures present in only part of the dataset are erroneously assigned to every sample[17,18]. To avoid this effect and accurately estimate the contribution of each mutational signature, we designed an algorithm—named *mmsig*—where the eight extracted mutational signatures in MM were fitted for each patient. The contribution of each mutational signature was corrected based on the cosine similarity between the original 96-class mutational profile and the reconstructed profile generated without that mutational signature ("Methods"). *mmsig* revealed SBS-MM1 activity only in samples obtained at relapse, from 9 patients (17%) (Fig. 1b, c and Supplementary Data 2). Interestingly, this mutational process was particularly enriched after exposure to high-dose melphalan and autologous stem cell transplantation (5/9 posttransplant vs. 1/29 relapsed after other therapies; Fisher's test $p = 0.001$). The only non-transplanted patient with evidence of SBS-MM1 activity (PD26414) had received several treatment lines, including melphalan, and cyclophosphamide for stem cell mobilization before the first sample collection. In contrast, SBS-MM1 was absent in all pre-treatment samples (i.e., SMM and newly diagnosed MM) (0/20 vs. 5/9; $p = 0.001$, Fisher's exact test). To further investigate association between treatment and SBS-MM1, we analyzed exome data from two independent validation cohorts: (1) 72 patients with paired samples collected at diagnosis and relapse, with 70% cases exposed to alkylating agents (CoMMpass trial); and (2) 40 MM patients all previously exposed to alkylating agents and refractory to both bortezomib and lenalidomide (Fig. 1d)[20,21]. In line with our observations, SBS-MM1 was not present in therapy-naive samples but was detected in the same patients at relapse and in the independent relapsed/refractory cohort. No clinical or recurrent genomic features were significantly associated with SBS-MM1, except prior exposure to melphalan (Supplementary Table 2).

Several therapy-related mutational signatures (e.g., platinum-SBS31/SBS35, azathioprine-SBS32, and SBS25) have been associated with transcriptional strand bias, reflecting transcription-coupled nucleotide excision repair activity on damaged DNA[9]. Similarly, SBS-MM1 showed a strong transcriptional strand bias in C>T, involving distinct trinucleotide (e.g., C[C>T]A, G[C>T]A, G[C>T]C, G[C>T]G, G[C>T]T) and five-nucleotide contexts (e.g., GG[C>T]AA, GG[C>T]AC, GG[C>T]AG, GG[C>T]CA, GG[C>T]TA, GG[C>T]TG), supporting the association between this mutational signature and chemotherapy-induced DNA damage (Fig. 1e and Supplementary Data 2–4).

A recent study investigated the mutagenic impact of 79 known or suspected environmental carcinogens on human-induced pluripotent stem cells in vitro[22]. Melphalan exposure was tested in this study; however, they excluded this experiment from further analysis due to a signal-to-noise-ratio <2[22]. Re-analyzing their published data by two independent de novo mutational signature extraction algorithms (hdp and SigProfiler), we identified a significant SBS-MM1 contribution in the melphalan-exposed cells (i.e., MSM0.10) but not in 15 control cell lines or 89 patients with chronic lymphocytic leukemia (CLL) (Supplementary Fig. 2). These negative controls were included in the de novo extraction to correct for the potential inter-bleeding of mutational signatures across different samples[18]. As a further confirmation, the 96-class mutational profile of MSM0.10 was best explained by the combination of SBS-MM1 and the "control" mutational signature, a signature detected in single-cell expansions irrespective of exposures and attributed to the experimental conditions (Supplementary Data 5 and 6). Finally, these results were confirmed by mutational signature fitting with *mmsig* (Supplementary Figs. 3 and 4).

Overall, these data suggest the existence of a distinct mutational signature associated with melphalan exposure, demonstrating that standard chemotherapy can alter the tumor genome in MM.

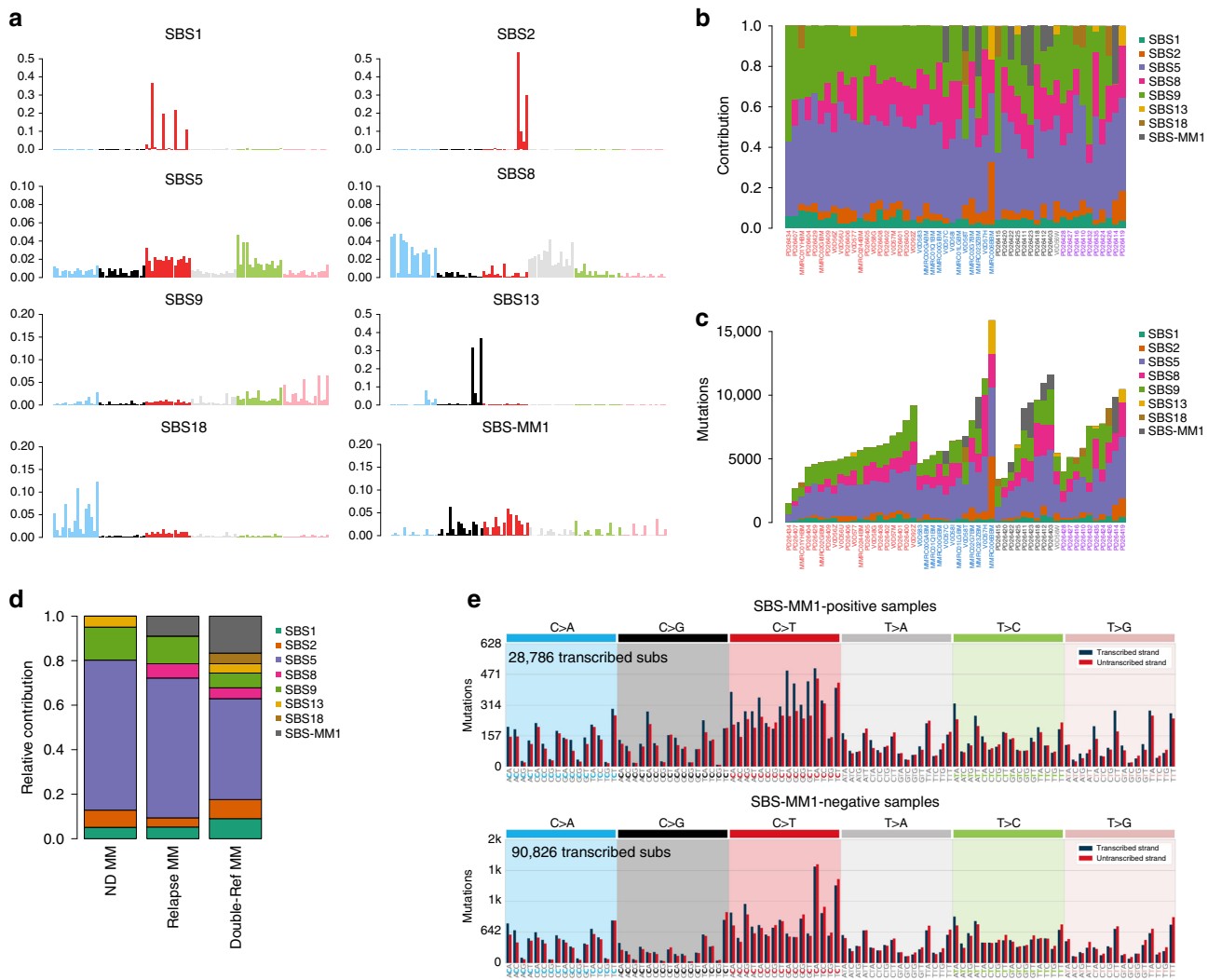

**Fig. 1 Multiple myeloma mutational signature landscape. a** The 8 mutational signatures extracted from 89 WGSs from 52 MM patients. **b, c** The relative (**b**) and absolute (**c**) contribution of each mutational signature for each patient. The x-axis labels are colored according to disease stage and treatment history: red = treatment naive; blue = relapsed cases without a full clinical annotation; black = relapsed cases after transplant with high-dose melphalan; purple = relapsed cases never exposed to transplant with high-dose melphalan; gray = missing clinical data. **d** Relative contribution of all eight MM mutational signatures on exome data collected at diagnosis (ND = newly diagnosed), relapsed, and when the disease was refractory to both bortezomib and lenalidomide (i.e., double-ref). **e** Transcriptional strand bias profile of patients with (top) and without (bottom) evidence of SBS-MM1.

Whether this contributes to the phenotype at relapse and subsequent evolutionary trajectory remains to be addressed.

Investigating the topography of mutational signatures involved in MM pathogenesis, we observed a strong enrichment of SBS9, SBS8, and SBS5 in late-replicating regions with low chromatin accessibility, in line with observations from other cancers (Supplementary Fig. 5)[23]. Similar patterns of enrichment were found for SBS-MM1.

Two clock-like processes (SBS1 and SBS5) were observed in all patients, in line with previous works (Fig. 1c, d)[6,7,9,17,24–26]. APOBEC (SBS2 and SBS13) was also confirmed as an essential mutational process, absent in only 9 patients (17%)[17,25,27,28]. While SBS2 was always the most common APOBEC signature (43/52; 83%), SBS13 activity was observed in 6 cases (11.5%) (Supplementary Data 2). Two of these were characterized by high mutation burden and >20% contribution from APOBEC (SBS2 +SBS13). Both patients had *IGH* translocations involving *MAFB* and *MAFA*, confirming the known association between these *IGH* translocations and high APOBEC mutational activity (Supplementary Data 1)[26,27]. Interestingly, these two cases were

the only ones without detectable non-canonical activation-induced deaminase (AID) (SBS9) activity.

APOBEC mutational activity in cancer can be sustained by two main isoforms: APOBEC3A and APOBEC3B. The contribution of each isoform can be estimated by considering the second base 5' to the mutated cytosine (4-nucleotide context)[29]. In solid cancers, APOBEC3A has been suggested as the most active isoform[9]. In contrast, we found a 1:1 ratio in all MM patients, with the notable exception of *MAFB/MAFA* translocated cases, where the ratio was higher (Fig. 2a). The association between *MAF* translocations and increased APOBEC3A activity over APOBEC3B was further confirmed in exome data from 692 newly diagnosed MMs with available *IGH* translocation data (Fig. 2b, c). Therefore, we asked whether variations in APOBEC3A/3B ratio in solid cancers could be similarly explained by the total APOBEC mutation burden. Analyzing the 4-nucleotide context of 788 WGSs from 27 tumor types with evidence of APOBEC activity enrolled within the PCAWG ICGC consortium, we observed a strong positive association between APOBEC3A/APOBEC3B ratio and APOBEC mutational burden (linear regression $p < 0.0001$; Fig. 2d, e).

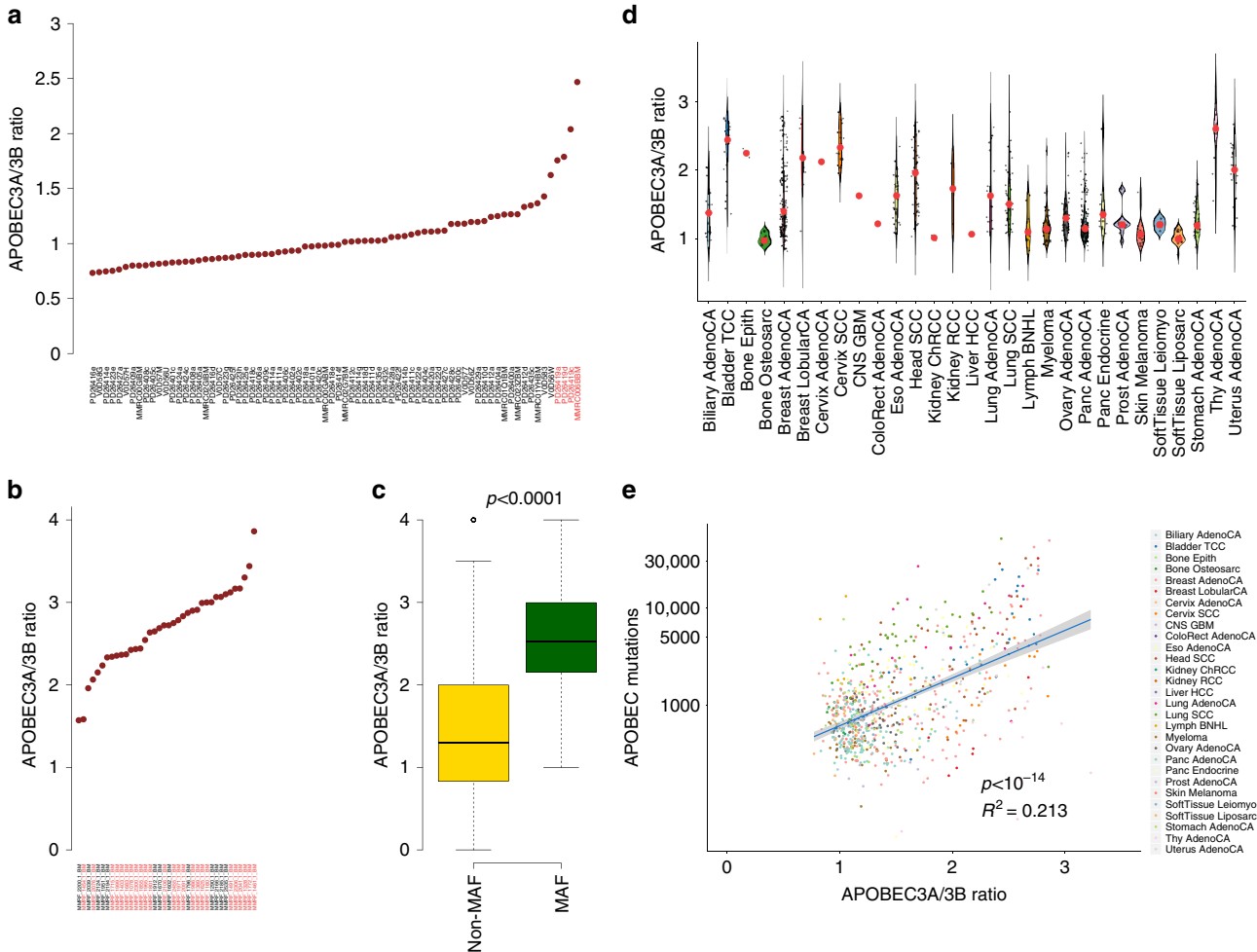

**Fig. 2 APOBEC3A and APOBEC3B activity in multiple myeloma and other cancers. a, b** The ratio between APOBEC3A/APOBEC3B in all multiple myeloma whole genomes (**a**) and in CoMMpass exomes with high APOBEC mutational burden and available structural variant data (**b**). Red and black x-labels were used to highlight patients with and without MAF translocations, respectively. **c** Differential APOBEC3A/APOBEC3B ratio between patients with ($n = 40$) and without MAF ($n = 652$) translocations ($p < 0.0001$ by *wilcoxon.text* R function). Boxplots show the median and interquartile range; observations outside this interval are shown as dots. **d** The APOBEC3A/APOBEC3B ratio in our MM cases and all tumors included in the PCAWG consortium with evidence of APOBEC activity. Red dots reflected the median APOBEC3A/3B ratio for each cancer type. **e** Correlation between APOBEC mutational burden (log scale) and APOBEC3A/APOBEC3B ratio. R-squared and p values ($p < 0.0001$) were estimated using linear regression (*lm* R function).

Conversely, the APOBEC 3A/3B ratio was closer to 1:1 when the global APOBEC mutation rate was low. Overall, this suggests the existence of two different APOBEC mutational patterns at the whole-genome level: one where a few hundreds of mutations are caused by both APOBEC3A and APOBEC3B and another where the APOBEC mutational burden is higher and mainly sustained by APOBEC3A. Tumors with high APOBEC activity showed similar patterns across MM and solid tumors, indicating a potential common mechanism.

**Kataegis and localized mutational events in MM.** Canonical and non-canonical AID activity (c-AID—SBS84 and nc-AID—SBS9, respectively) has been described in kataegis regions from immunoglobulin loci in MM and other post-germinal center lymphoproliferative disorders[9,17,24,30,31]. c-AID activity was not extracted genome-wide by SigProfiler or hdp, consistent with its localized mutational activity on specific hotspots such as the immunoglobulin loci (Supplementary Fig. 6)[18]. Here both processes act in clusters of mutations, reflecting one of the most important germinal center (GC) processes: somatic hypermutation (SHM)[32]. Outside of the immunoglobulin loci,

127 kataegis events were extracted (median 2 per patient; range 0–13), mostly induced by nc-AID or APOBEC (Fig. 3a–d). Forty-seven non-immunoglobulin kataegis (37%) were located <1 Mb from at least one structural variant (SV) breakpoint. Interestingly, 95% of these events were caused by APOBEC (Fig. 3c–e). In contrast, nc-AID kataegis events were usually not associated with SVs (Fig. 3d–f). Overall, this suggests two main routes of acquisition of kataegis outside of the immunoglobulin loci in MM: one caused by AID activity, likely reflecting GC exposure; and another caused by APOBEC, usually co-occurring with structural chromosomal changes.

**Temporal patterns of MM mutational signatures.** To investigate the activity over time of each mutational process, we reconstructed the phylogenetic tree for each patient with more than one sample collected at different time points ($n = 26$) and subdivided all mutations into three main categories[15,17]: (1) early clonal (i.e., mutations identified as clonal in all samples); (2) late clonal (i.e., clonal in at least one sample but not in all); and (3) subclonal (not clonal in any sample) (Fig. 4a). nc-AID was strongly enriched among early clonal mutations compared to late

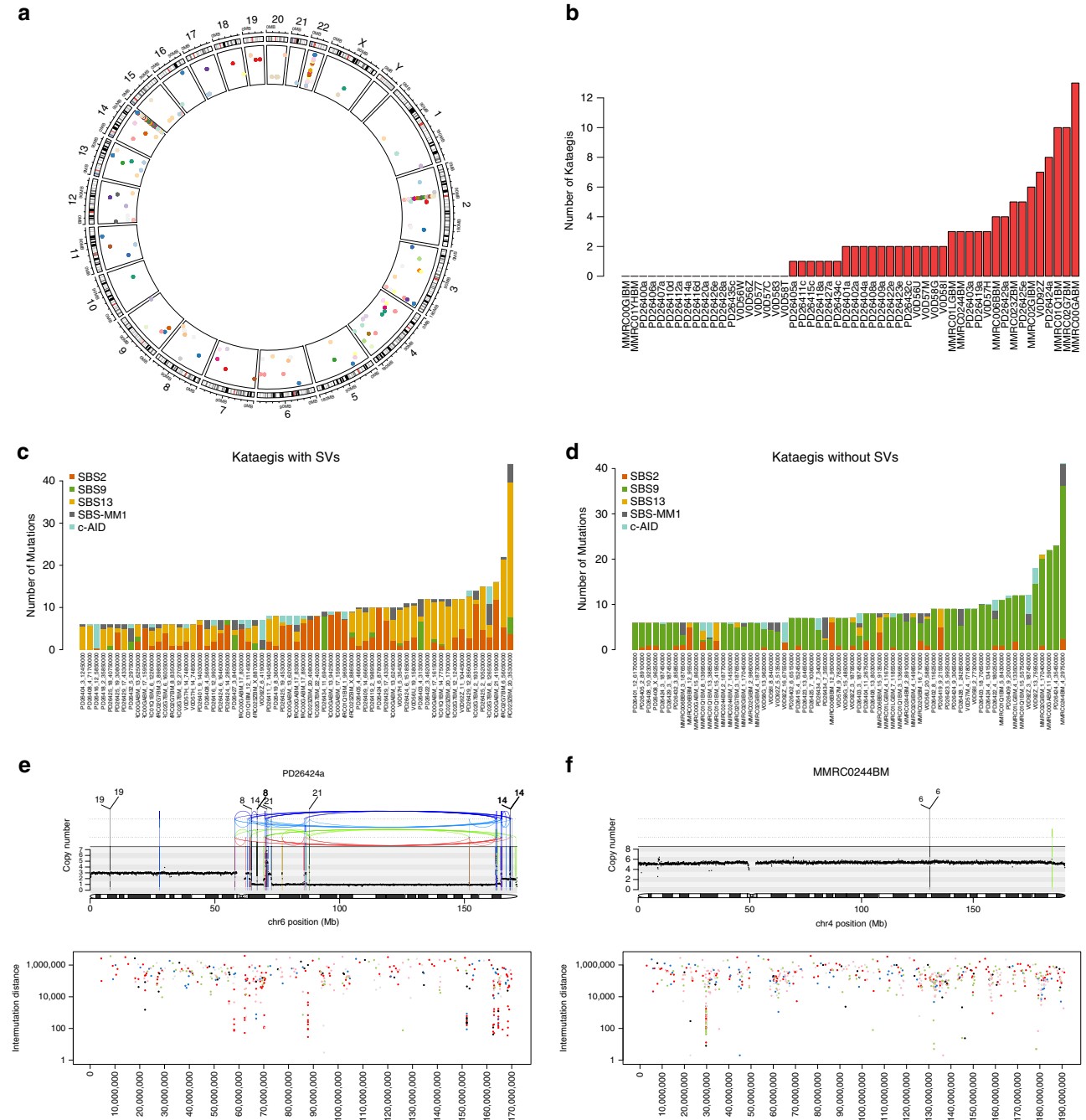

**Fig. 3 Kataegis in multiple myeloma. a** The distribution of all kataegis events extracted across 52 MM patients, each color reflecting a distinct patient. **b** Number of kataegis events per patient. **c** In kataegis events close to at least one SV breakpoint, APOBEC (SBS2 and SBS13) was the main mutational process. **d** Other kataegis events not associated with any SV breakpoints were dominated by nc-AID. **e** Example of APOBEC-mediated kataegis associated with chromothripsis. **f** Example of nc-AID kataegis not associated with SV. In **e**, **f**, black dots represent the chromosome ploidy status. Vertical black, blue, green, and red lines reflect translocations, inversions, tandem duplications, and deletion, respectively. Below each copy number/structural variant plot, we reported the inter-mutational distance of all SNVs, color-coded by class (blue: C>A, black C>G, red C>T, gray T>A, green T>C, pink T>G).

clonal and subclonal (p < 0.0001; pairwise Wilcoxon test). In 18 cases (69%), nc-AID was exclusively detected among early clonal mutations. Interestingly, a persistent nc-AID contribution was still observed among late clonal and subclonal mutations in 8 (31%) patients, suggesting GC exposure and AID-driven subclonal diversification after the most recent common ancestor (MRCA) (Fig. 4a).

We went on to look for additional evidence of AID activity after the MRCA, either for a limited period of time or ongoing at the time of MM diagnosis. First, we searched for the mutational

footprint of c-AID activity in the *IGH* locus. In 26 paired WGS samples, we identified 196 unique mutations that were not shared across all samples from the same patient, showing clear evidence of c-AID contribution (Supplementary Fig. 7A). These results were confirmed in sequential samples from 72 patients in CoMMpass study by whole-exome sequencing (Supplementary Fig. 7B, C). Next, we reconstructed the third complementarity determining region (CDR3) of *IGH* from WGS data in the same 26 paired samples (Supplementary Table 3). Interestingly, the CDR3 sequences were identical across sample pairs in all patients.

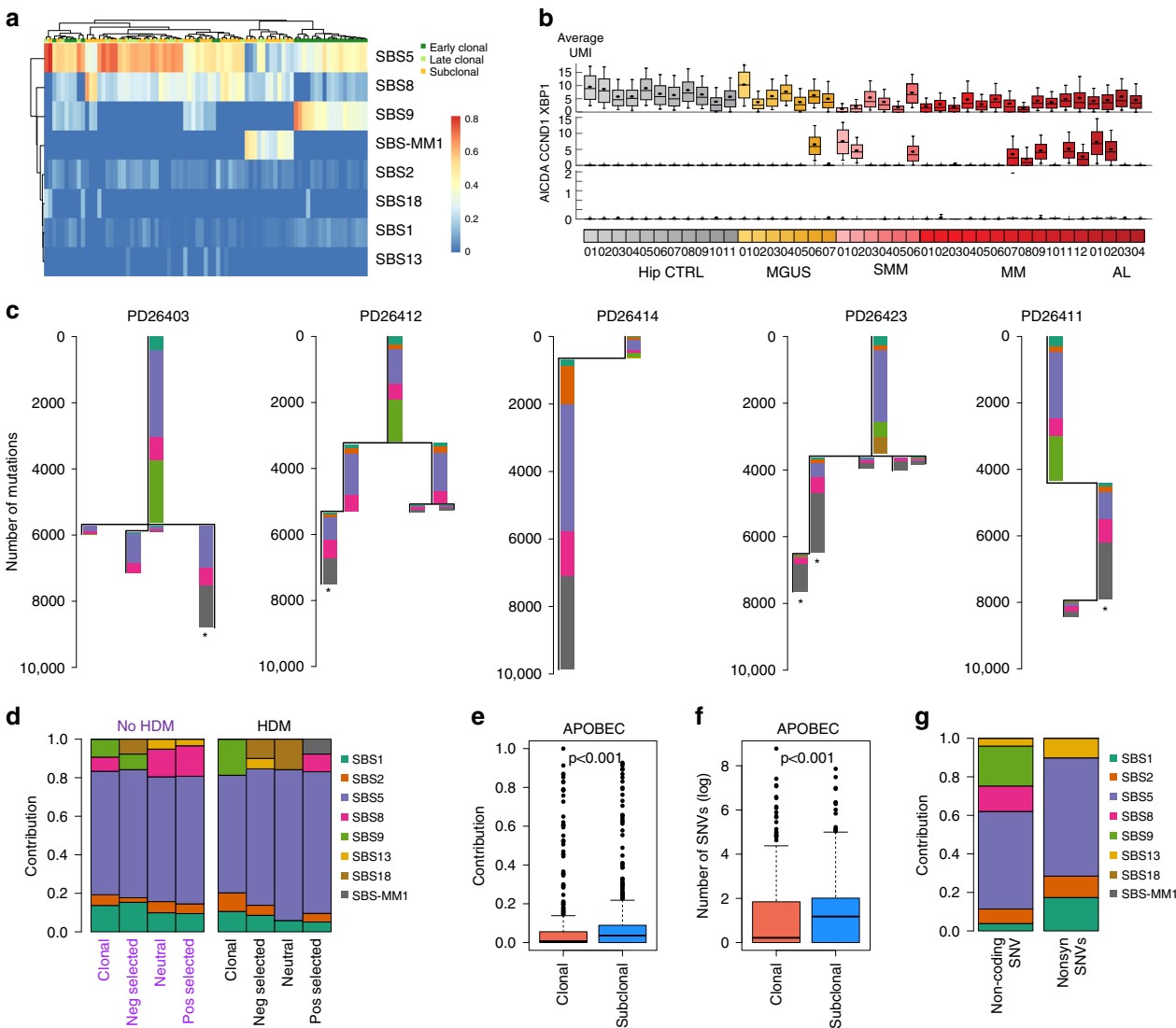

**Fig. 4 Chronological reconstruction of mutational processes in multiple myeloma. a** Heatmap showing the relative contribution of mutational signatures during the three evolutionary phases: early clonal, late clonal, and subclonal. Only patients with more than one sample collected at different time points were considered (n = 26). **b** Boxplots of single-cell gene expression for specific genes across 20,586 single bone marrow plasma cells from 29 newly diagnosed patients and 11 control donors. Each box represents 0.25–0.75 percentile of unique molecular identifier (UMI) count with line extension to 0.1–0.9 percentile; dot represents the mean UMI count. Patients are color-coded by disease (gray—controls from hip replacement surgery (Hip CTRL), yellow—MGUS, light red—SMM, red—MM, dark red—AL amyloidosis). **c** Mutational signature composition of each phylogenetic tree of patients with multiple samples collected at different time points and evidence of SBS-MM1. Asterisks represent the presence of transcriptional strand bias in SBS-MM1 single-, tri- and penta-nucleotide contexts. **d** Mutational signature contributions in CoMMpass patients with samples collected at baseline and first relapse and investigated by both whole exome and low coverage long insert WGS (n = 24). Mutations were grouped according to their position in each patient phylogenetic tree. x-axis labels colored by transplant and high-dose melphalan status: black for high-dose melphalan (HDM) exposed and purple for non-exposed. **e**, **f** The relative (**e**) and absolute (**f**) contribution of APOBEC (SBS2 and SBS13) to clonal and subclonal mutations from 788 CoMMpass MM patients. p Values (p < 0.001) were calculated using the wilcoxon.text R function. **g** Mutational signature contributions for non-coding compared with nonsynonymous (Nonsyn) mutations in newly diagnosed MM patients from both CoMMpass and WGS data.

Highly conserved CDR3 sequences over time in MM is consistent with previous studies[33], including CDR3 reconstruction from RNA sequencing of sequential samples in the CoMMpass cohort[34].

To determine whether AID mutational activity is still ongoing at the time of MM diagnosis, we analyzed expression of the *AICDA* gene (encoding AID). There was no evidence of *AICDA* expression in bulk RNA sequencing (n = 792 in the CoMMpass study; Supplementary Fig. 7D) or at the single-cell level (Fig. 4b), consistent with previous reports that AID protein is not expressed in MM cells[35]. Furthermore, single-cell analysis of

immunoglobulin CDR3 sequences revealed no intra-clonal variation[36]. Taken together, these observations suggest that AID mutational activity persists for some time after disease initiation, driving late clonal and subclonal mutagenesis in a subset of patients, but is no longer active at the time of MM diagnosis. We hypothesize that myeloma precursor cells may be antigen responsive, leading to repeated cycles of GC re-entry until the disease becomes antigen and GC independent.

SBS-MM1 and its characteristic pattern of transcriptional strand bias were detected only among late clonal and subclonal mutations in patients exposed to melphalan, consistent with an

association with treatment (Fig. 4c and Supplementary Fig. 8). Patient PD26403 was an emblematic example of the changing mutational signature landscape over time in relation to disease progression and treatment (Supplementary Fig. 9). This patient was the only one with samples collected during the smoldering phase, at the time of progression into symptomatic MM, and at first relapse after therapy with high-dose melphalan. Both progressions were characterized by a branching evolution, but SBS-MM1 was only detected in mutations selected after therapy, not after spontaneous progression from SMM to MM. To further investigate the association between melphalan exposure and SBS-MM1, we analyzed the mutational signature landscape of 24 patients enrolled in the CoMMpass trial with whole exome and low-coverage WGS performed at baseline and the first relapse (Supplementary Data 7). Reconstructing the phylogenetic tree of each patient, we divided mutations into four main groups: clonal (i.e., trunk of the phylogenetic tree) and those following one of the three selection patterns from diagnosis to relapse: negative, positive, and neutral (i.e., unchanged cancer cell fraction) ("Methods"). Each group was further separated by exposure to high-dose melphalan. In line with its therapy-related etiology, SBS-MM1 was identified only among mutations positively selected after melphalan exposure (Fig. 4d).

In contrast to AID, APOBEC was significantly enriched among late clonal and subclonal mutations ($p < 0.0001$; pairwise Wilcoxon test), with 8 patients (31%) having no detectable APOBEC among early clonal variants (Fig. 4a). The increasing activity of APOBECs during late phases of cancer development was also confirmed in the CoMMpass exome cohort (Fig. 4e, f; $p < 0.001$ Wilcoxon test). Interestingly, in this large exome series, SBS2 and SBS13 were particularly enriched among nonsynonymous compared to non-coding mutations (21.3% vs. 11.6%; Fisher's test $p < 0.001$), confirming the role of APOBECs in MM progression and subclonal diversification (Fig. 4g)[37]. SBS8 and SBS18 were detected in 45 (86%) and 6 (11%) cases, respectively, usually with higher contribution among late clonal and subclonal mutations.

The early clonal clusters usually accounted for >50% of the total mutation burden (median 70.6%; range 33–100%), and its composition reflects the sum of different mutational processes active during different time windows in the preclinical phase. To investigate the temporal activity of each mutational process among early clonal mutations, we took advantage of primary and secondary chromosomal gains in 33 patients where early clonal mutations could be divided into duplicated mutations (i.e., present on two alleles and therefore acquired before the duplication) or non-duplicated mutations (i.e., detected on a single allele), reflecting either pre and post-gain mutations on the minor allele or post-gain mutations acquired on one of the duplicated alleles ("Methods")[15]. While SBS1 and SBS5 were active in both groups of mutations (pre- and post-gain), nc-AID (SBS9) was significantly enriched in the pre-gain mutations ($p < 0.0001$), highlighting its involvement in the earliest phase of MM pathogenesis (Fig. 5a, b). Mutations attributed to nc-AID were also detected on one single allele. This can be explained by either persistent AID activity causing mutations after the duplication or mutations acquired before the gain on the minor allele. To resolve this ambiguity, we restricted our analysis to large areas of copy neutral loss of heterozygosity (LOH) in 16 patients, where the minor allele was lost and the major duplicated. In these cases, all mutations present on a single allele must be acquired after the gain. While nc-AID activity was more predominant before the LOH event, it could still be detected after (Fig. 5c). Similarly, we characterized mutations acquired between two independent gains of the same allele (i.e., tetrasomy). Sixteen clonal tetrasomies were observed in 10 patients. Using our molecular time approach, we

estimated that six patients had one or more tetrasomy acquired in two distinct time windows[2,15]. In three of these, nc-AID was identified between the two duplications (i.e., mutations on two alleles out of four) (Fig. 5d). Notably, out of the 12 patients without subclonal nc-AID mutations, 2 patients with paired samples showed evidence of prolonged nc-AID activity after the first gain. Overall, these data further support the notion that AID mutational activity is not confined to the initial GC contact when transformation occurs but in most cases continues to shape the genome across a considerable time span.

Interestingly, APOBEC activity was generally low ($n = 8$) or absent ($n = 20$) before chromosomal gains except for the two MAFA/MAFB MM patients. Here APOBEC was highly active already in the earliest phases, and nc-AID was not detectable at any stage. To investigate whether AID was active at all in these two cases, we analyzed the mutational profiles of their immunoglobulin loci. Despite the limited number of mutations, a minimal c-AID activity was detected (Supplementary Fig. 10A). To further confirm c-AID activity on the immunoglobulin loci in MM patients with high APOBEC mutational burden, we interrogated exome data from 70 and 863 newly diagnosed MM patients with high and low APOBEC activity, respectively[26]. No differences were observed between these two groups (Supplementary Fig. 10B, C), confirming that all cases of MM have undergone SHM.

Although SBS5 causes the majority of mutations during all phases of tumor development, different temporal patterns of AID and APOBEC mutagenesis may represent footprints of cellular processes and exposures during the evolutionary trajectory toward MM. Our data suggest at least four such patterns (Fig. 5e): (1) where APOBEC plays a major role during all evolutionary phases, including cancer initiation, and where AID contribution is minimal; (2) where APOBEC is not involved in any phase; (3) where AID is active only during the very early phase of cancer initiation; and (4) where AID is the main mutational process since the very first phases of cancer development and still plays a role in the late subclonal diversification.

**Clock-like mutational process in MM.** Similar to other cancers[2,6,9], we found SBS1 and SBS5 to be active in all phases of MM in all patients. In many cancers and normal tissues, these mutational signatures are correlated with patient age[4,6–8,38,39]. Previous results from a small cohort of MM exomes ($n = 68$) showed a modest correlation for SBS1 but not for SBS5[6]. Here we employed several approaches to determine whether MM does indeed have a clock-like mutational process, integrating our WGS cohort with a large exome series.

First, we assessed the relationship between patient age and the mutational burden attributed to each mutational signature at diagnosis of MM in the CoMMpass exome dataset ($n = 764$) using linear regression models. Significant relationships were identified for SBS5 [slope 1.55 (95% confidence interval (CI) 0.97–2.14); $p < 0.0001$] and SBS1 [slope 0.29 (95% CI 0.13–0.45); $p = 0.0003$], but none of the other mutational signatures (Fig. 6a and Supplementary Data 8). Despite the strong correlation, there was considerable variation in SBS1 and SBS5 mutational burden between individual patients that could not be explained by age alone. Next, we applied linear mixed-effects (LME) models to 72 patients in the CoMMpass dataset with paired diagnosis and relapse exomes, allowing for each patient to accumulate mutations at a different rate[5]. The LME model confirmed a strong linear relationship between SBS5 and age ($p < 0.0001$) and a weaker association for SBS1 (Fig. 6b and Supplementary Data 8). Finally, to address the SBS5 mutation rate over time in more detail, we applied similar LME models to our WGS cohort

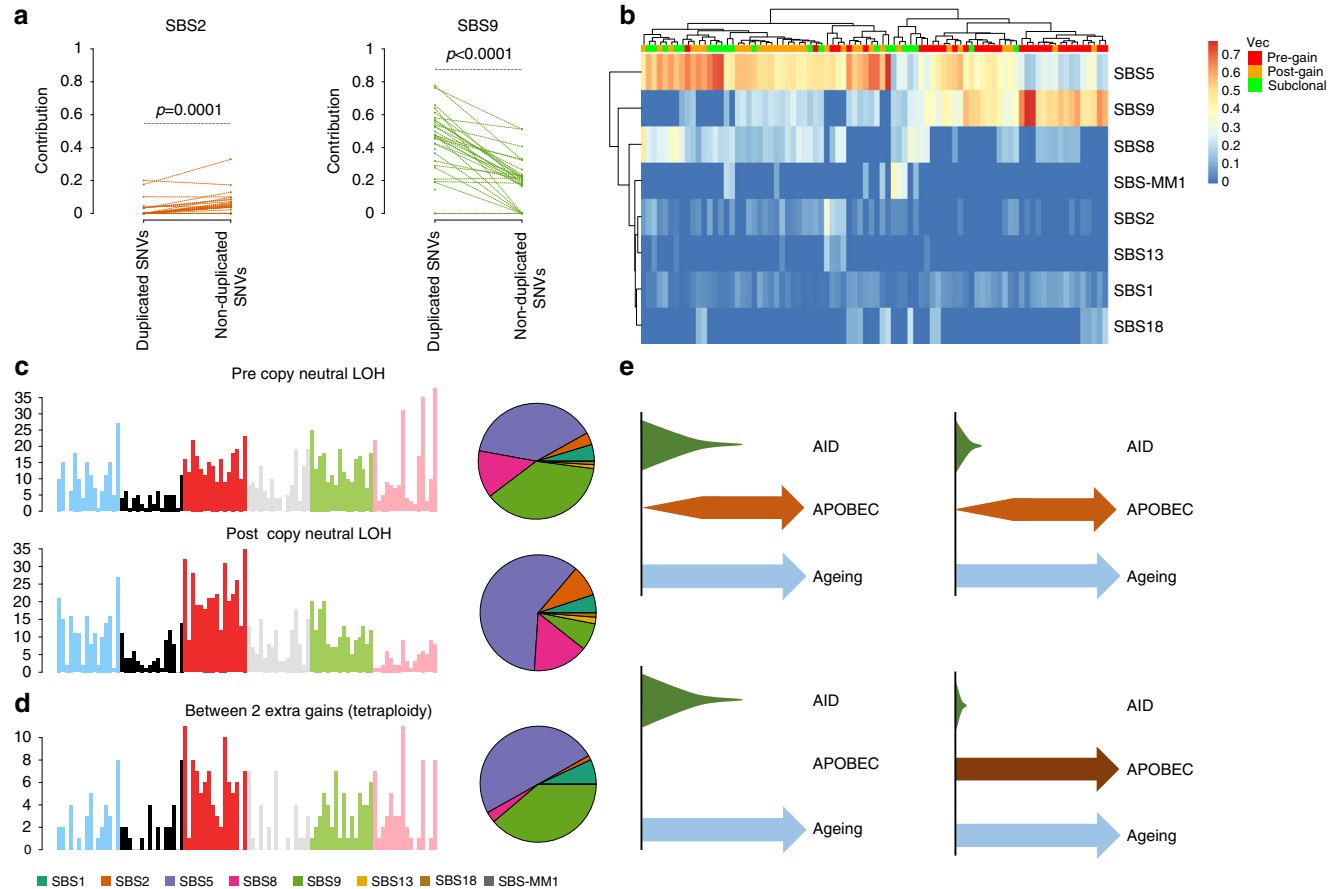

**Fig. 5 AID and APOBEC activity during early phases of cancer development. a** Relative contribution of SBS2 and SBS9 before and after large chromosomal gains. Each patient's clonal mutation catalog was divided into two groups: mutations acquired before the gain (VAF corrected for the normal contamination = 66%), mutations present on one single allele (VAF corrected for the normal contamination 33%). *p* Values were estimated with the *wilcoxon.text* R function. **b** Heatmap showing the relative contribution of all mutational signatures across different time windows for each patient. **c** 96-class mutational profile of all mutations acquired before (top) and after the copy neutral LOH (bottom). **d** 96-class mutational profile of all mutations acquired on one duplicated allele before its second duplication. **e** Schematic representation of the four main mutational signature patterns over time. In the bottom right panel, APOBEC is colored in brown to reflect the high APOBEC contribution.

(the entire computational workflow is reported in Supplementary Data 9). Here, thanks to the higher resolution of WGS data, we were able to quantify the SBS5 mutational activity in each subclone and define the major evolutionary trajectories ("Methods")[5]. This is required because mutations are independently acquired across distinct subclones. Collapsing all subclones together would, therefore, overestimate the mutation rate. Again, we observed that the SBS5 mutation burden increased linearly with patient age, at an average rate of 38.8 mutations per year across the cohort (95% CI 35.74–41.78), with a between-patient standard deviation of 7.1. (Fig. 6c, Supplementary Fig. 11, and Supplementary Data 9). The between-patient variation in our study was similar in magnitude to that observed by Mitchell et al. in kidney cancer using a similar approach, reporting an average mutation rate of 89 per year, with a between-patient standard deviation of 17[5].

Several factors may affect the power of WGS to identify mutations in a tumor sample, such as the sequencing coverage, tumor purity, and ploidy. If these factors confound the estimation of SBS5 mutation burden, adjusting for them in an LME model may provide a more accurate molecular clock. Furthermore, it is possible that the SBS5 mutation rate accelerates during the late phases of tumor evolution and subclonal diversification. We tested for this effect by including a quadratic term for age in our

LME model, effectively allowing the mutation rate to parabolically increase with age[5]. Adjusting for the above features did not improve LME model fit, and the estimated SBS5 mutation rate remained largely unchanged (Supplementary Fig. 12, Supplementary Data 9) Furthermore, the estimated SBS5 mutation rate was independent from disease stage (i.e., SMM, newly diagnosed MM, or relapsed MM; Bonferroni–Holm adjusted *p* = 1 by pairwise Wilcoxon test; Supplementary Fig. 11, Supplementary Data 9). Taken together, our data suggest that SBS5 mutations accumulate at a constant rate in each MM tumor, similar to other cancers and normal tissues[4–7,38,39].

Considering the clock-like behavior of SBS5 in MM, we would expect a similar pattern in other B cell-derived lymphoproliferative disorders. To confirm this, we interrogated 89 CLL and 107 B cell lymphoma genomes included in the ICGC/PCAWG consortium. The association between SBS5 mutational burden and patient age was confirmed by linear regression analysis in both non-hypermutated B cell lymphomas [slope 28.21 (standard error 4.87); *p* < 0.0001] and CLL [slope 18.6 (standard error 4.8); *p* = 0.0006] (Fig. 6d and "Methods"). Interestingly, mutated and unmutated CLL showed a similar slope, suggesting that the underlying biology is more important in determining the SBS5 mutation rate over time than subsequent environmental exposures (e.g., antigen-driven selection in the GC).

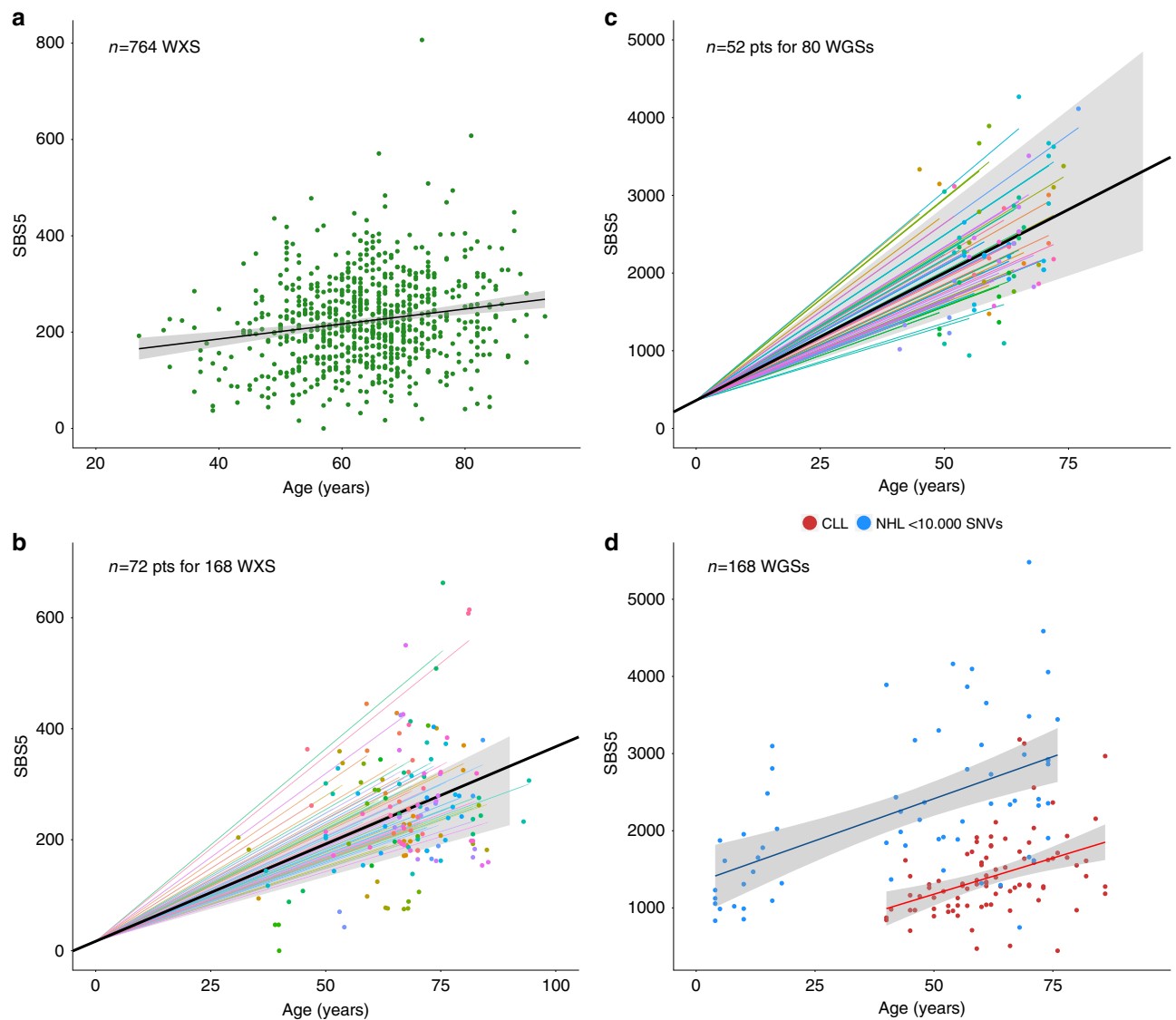

**Fig. 6 Clock-like properties of SBS5 in multiple myeloma. a** Linear regression model ($p < 0.0001$ with *lm* R function) for SBS5 mutational burden and age in 764 CoMMpass whole-exome sequencing cases. **b** Linear mixed-effects model for SBS5 mutation rate in 72 CoMMpass patients with sequential samples collected at different time points ($p < 0.0001$ with *lmer* R function). Points represent observed SBS5 mutational burden in phylogenetic branches, colored by patient. Colored lines represent patient-specific SBS5 mutation rates (i.e., slopes), with the population average as a black line surrounded by a shaded 95% confidence interval. **c** Linear mixed-effects model for SBS5 mutation rate in our cohort of multiple myeloma genomes; legend as in **b** ($p < 0.0001$ with *lmer* R function). **d** Linear regression model for SBS5 mutational burden and age in non-hypermutated B cell lymphomas and CLL WGSs included in the PCAWG consortium ($p < 0.0001$ with *lm* R function).

**Timing the initiation and evolution of MM.** Having established SBS5 as a molecular clock in MM, we applied it to estimate the absolute age of each patient when landmark events occurred during tumor evolution[2,4,5]. This included primary and secondary chromosomal gains as well as emergence of the MRCA. Based on the SBS5 mutation rate per year of each patient, we translated the number of SBS5 mutations that had accumulated before each landmark event into an estimated patient age when that event occurred. MRCA timing was estimated based on early clonal SBS5 mutation burden for patients with multiple samples available. The median time lag between the age at MRCA and at the first sample collection varied greatly (14.2 years; range 0–45) (Supplementary Fig. 13 and Supplementary Data 9). In some patients, the estimated age when the MRCA emerged was very close to the sample collection, which could also reflect an overestimation of clonal mutation burden. This may occur when a branch of the tumor phylogenetic tree is inappropriately assigned

as the trunk because of missed spatial and genomic heterogeneity and sampling bias[40].

To estimate the patient age when the first MM driver event was acquired, we leveraged the high prevalence of chromosomal gains in MM and their involvement in cancer initiation. Twenty-eight patients (54%) had at least one multi-gain event, and in six of these patients, there was evidence of gains acquired in two distinct time windows (Fig. 7, "Methods," and Supplementary Data 9)[2,15]. Using the patient-specific SBS5 mutation rate, we estimated the patient age when these events occurred, similar to what was done for the MRCA. Intriguingly, the first chromosomal duplication was acquired on average 37.5 years (range 8–66) before sample collection. In 21/27 (78%) cases, the first multi-gain event occurred before 30 years of age and in 12/27 (44%) before 20 years (Fig. 7b and Supplementary Data 9). Overall, the second multi-gain event tended to occur closer to the first sample collection (median 8, range 0–37), reflecting a protracted

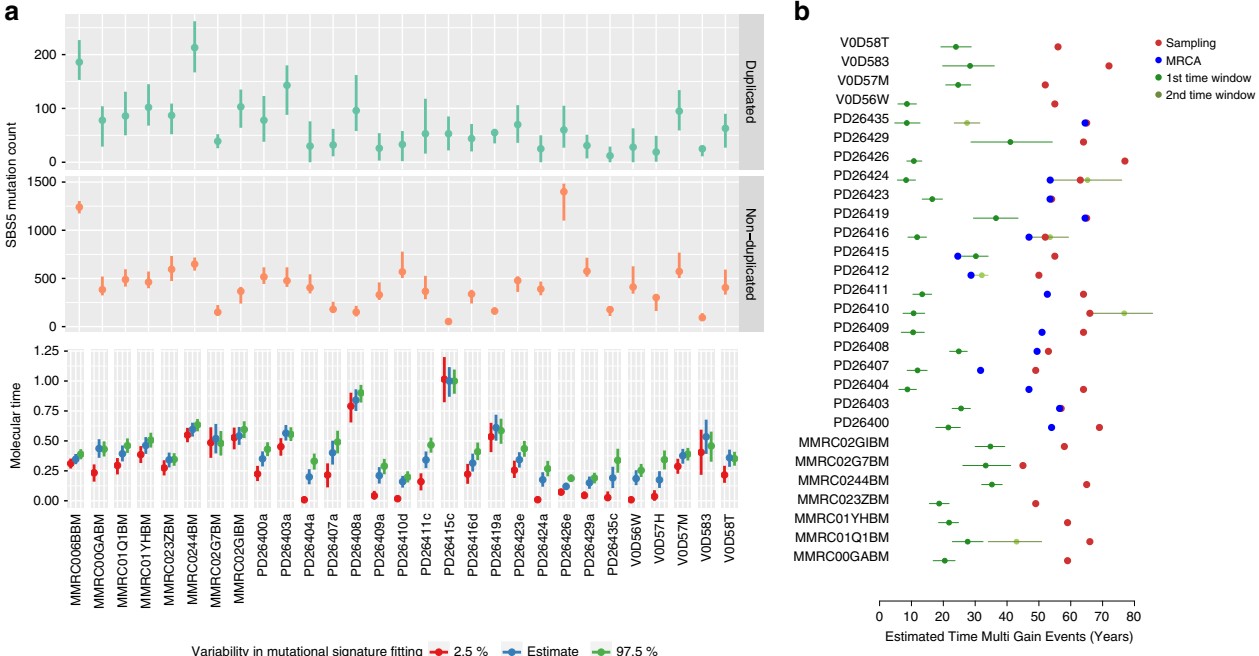

**Fig. 7 Timing the first multi-gain event in multiple myeloma. a** SBS5 mutation burden on duplicated (CN = 2) and non-duplicated (CN = 1) alleles of large chromosomal gains occurring in the same time window (top), and molecular time estimates based on the corrected ratio of duplicated and non-duplicated mutations (dots below). 95% confidence intervals in the upper panel are based on the uncertainty of mutational signature fitting, which are propagated to different estimates (colors) in the panel at the bottom. 95% confidence interval bars in the lower panel represent the uncertainty of molecular time estimation given a number of duplicated and non-duplicated SBS5 mutations. **b** Estimated patient age at the first (dark green) and second (green) multi-gain events with 95% CIs. Blue and red dots represent age at the MRCA emergence and first sampling, respectively. MRCA timing is only shown for patients with multiple samples.

evolutionary process where drivers accumulate and undergo selection over time. Finally, we tested the robustness of our absolute timing estimates to uncertainty in mutational signature fitting ("Methods"). This confirmed our conclusion that the first multi-gain event occurs before 30 years of age in the majority of patients (Supplementary Data 9, Supplementary Fig. 14).

## Discussion

In this study, we drew on large WGS and whole-exome sequencing datasets to reconstruct the timeline of mutational processes shaping the MM genome, providing new insights into its origin and evolutionary trajectories.

We identified a mutational signature related to melphalan exposure in relapsed MM, indicating a mutagenic effect of this agent in clonal plasma cells. This is important given that high-dose melphalan is considered to be part of the standard first-line treatment in eligible patients[41]. Interestingly, not all patients exposed to melphalan showed evidence of SBS-MM1. We propose two potential explanations for this observation. First, according to the recently proposed single-cell expansion model, cells acquiring melphalan-induced mutations must undergo clonal expansion for those mutations to be detectable by bulk WGS[42]. Conversely, in a setting of neutral evolution[25], where no single cell has a major growth advantage, newly acquired mutations would not reach the threshold for detection by WGS (Fig. 8a). Second, post-melphalan relapse may arise from tumor cells contaminating the autologous stem cell graft[43]. Autologous stem cells are collected before high-dose melphalan therapy and re-infused after and thus will not have been exposed to melphalan (Fig. 8a). Future studies will reveal whether mutations induced by melphalan shape the evolution of relapsed disease or development of secondary malignancies, such as acute myeloid leukemia[44].

Mutational processes related to AID and APOBEC activity displayed at least four distinct temporal patterns during MM pathogenesis. Interestingly, our data indicate that AID activity is not limited to the first GC reaction but persists in at least a subset of patients, potentially affecting disease evolution. This suggests that pre-malignant MM cells behave similar to memory B cells, capable of re-entering the GC to undergo clonal expansion decades before MM diagnosis (Fig. 8b)[45,46]. Indeed, the notion of a prolonged antigen-responsive phase in MM is supported by studies of lipid-reactive gammopathies, where antigen stimulation resulted in clonal plasma cell expansion and M spike increase in mice[47,48].

Based on patient-specific estimates of SBS5 mutation rate and inferred molecular time, we estimated that the first chromosomal duplication in MM patients was usually acquired before 30 years of age. Despite uncertainty in our estimates, similar to previous reports[4,6–8,38], we observed a clear pattern across patients suggesting early acquisition of gains. Furthermore, our findings are in line with epidemiological data. Large population-based studies have shown that MGUS starts to become detectable at 30 years of age, after which the prevalence continues to increase[49,50]. Further narrowing the window between initiation and a clinically detectable entity, emerging studies using highly sensitive mass spectrometry assays have identified monoclonal proteins in serum up to 10 years before becoming detectable by conventional methods[51].

Our analysis provides a glimpse into the early stages of myelomagenesis, where acquisition of the first key drivers precedes cancer diagnosis by decades. Defining the time window when transformation occurs opens up for new avenues of research: to identify causal mechanisms of disease initiation and evolution, to better define the optimal time to start therapy, and ultimately develop early prevention strategies.

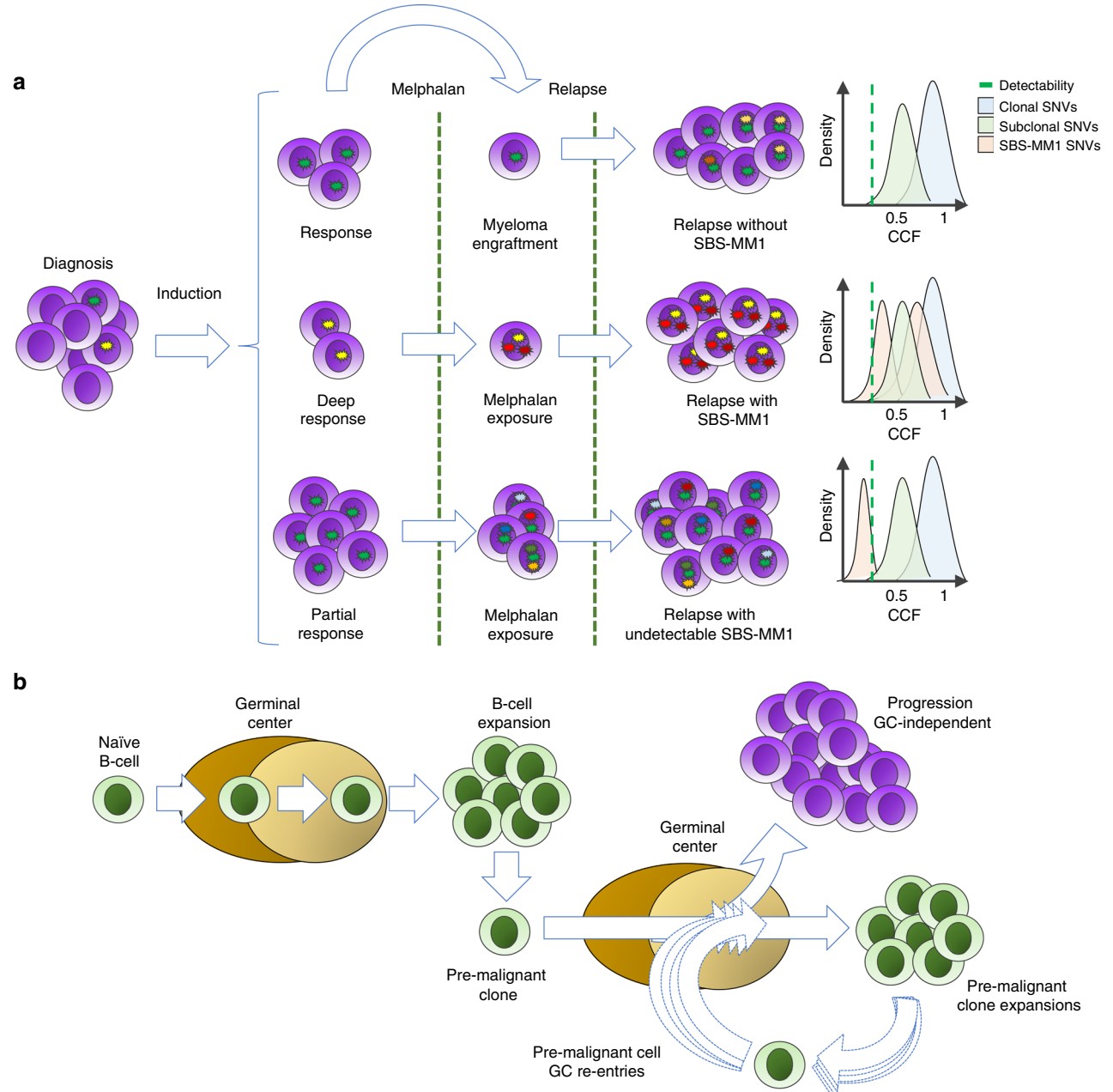

**Fig. 8 Single-cell expansion model for melphalan and GC-related mutational signatures. a** Cartoon summarizing the mutagenic impact of melphalan in MM patients. In the case of tumor cell post-transplant engraftment, MM cells will not have any melphalan-induced mutations (top). When a cancer cell is exposed to melphalan, it acquires unique mutations that will be detectable only in case of a single tumor cell expansion (center). In the absence of single-cell expansion, melphalan-induced mutations will be present in each exposed tumor cell but undetectable due to the low cancer cell fraction. **b** Proposed model of GC-dependent clonal expansion and AID-mediated hypermutation during the pre-malignant phase of MM development. Finally, a single cell becomes independent of the GC, going on to expand and give rise to the MM clone.

## Methods

**Subject details**. The study involved the use of human samples, which were collected after written informed consent was obtained (WTSI protocol number 15/046)[15]. Samples and data were obtained and managed in accordance with the Declaration of Helsinki. DNA was extracted from CD138+ cells purified from the bone marrows of 30 patients. Twenty-six (86%) patients had more than one sample collected at different time points for a total of 67 tumor samples and 30 matched normals collected from peripheral blood mononuclear cells (Supplementary Table 1). Samples were collected at different clinical time points: SMM ($n = 11$), newly diagnosed MM ($n = 15$), and relapsed MM ($n = 41$) (Supplementary Table 1).

**Whole-genome sequencing**. Short insert 500-bp genomic libraries were constructed, flowcells were prepared, and sequencing clusters were generated

according to Illumina protocols. We performed 108 base/100 base (genomic) paired-end sequencing on HiSeq X10 genome analyzers. The average sequence coverage was 38.7-fold. Short insert paired-end reads were aligned to the reference human genome (GRCh37) using Burrows–Wheeler Aligner, BWA (v0.5.9). Sequencing data have been deposited in the European Genome phenome Archive (EGA) under accession number EGAD00001001898.

To this cohort, we added 22 MM patients with available WGS data (dbGap: phs000348.v2.p1). Raw data was downloaded and processed similar to the in-house samples.

**Whole-genome analysis pipeline**. All 89 MM samples from the 2 cohorts were uniformly analyzed at the Wellcome Sanger Institute[15]. Briefly, deduplicated aligned BAM files were analyzed using the following published tools: (1) ASCAT and Battenberg for clonal and subclonal copy number variants; (2) BRASS for SVs;

(3) Caveman for single-nucleotide variants (SNVs); (4) Pindel for small insertions–deletions; and (5) the Dirichlet process to determine the tumor clonal architecture. Phylogenetic tree reconstruction was performed applying the pigeonhole principle to mutational clusters defined by the Dirichlet process[15]. The MRCA was defined as the cluster of mutations detected as clonal in all samplers (i.e., trunk of the phylogenetic tree).

Kataegis, or foci of localized hypermutation, was defined as ≥6 consecutive mutations with an average inter-mutation distance of ≤1 Kb.

**Whole-exome sequencing and analysis.** Two cohorts of exome data were included in this study. One comprised of 933 MM patients enrolled within the CoMMpass trial (NCT01454297; phs000748.v1.p1). The CoMMpass data were generated as part of the Multiple Myeloma Research Foundation Personalized Medicine Initiative (https://research.themmrf.org). PhyloWGS[52] was used to reconstruct the phylogenetic tree structure from whole-exome sequencing of 788 patients with available copy number data from low coverage long-insert WGS (median 4–8×). Seventy-two patients had a sample collected both at baseline and at first relapse where whole exome sequencing was performed; 24 of these patients also had low-coverage long-insert WGS data (Supplementary Data 7). For the latter 24 patients, mutations were divided into (i) mutations that were clonal in both samples (trunk of the phylogenetic tree); (ii) mutations that were either positively or negatively selected at relapse; and (iii) subclonal mutations that did not show any difference in cancer cell fraction from diagnosis to relapse (neutral).

The second cohort was comprised by 40 MM patients refractory to both immunomodulatory agents and proteasome inhibitors (INT 15/14; EGAS00001003709)[20,21]. Tumor (CD138+ bone marrow plasma cells) and matched germline samples (DNA from buccal swabs) were processed from 40 patients (80 total samples). DNA was enriched for the coding fraction of the genome using the SureSelect All Exon V6 Kit (Agilent Technologies) and the NextSeq500 machine (Illumina) was used to sequence an average of 15.63 Gb per patient on a 75-bp paired-end protocol. Samples were sequenced to an average depth of 97×. Paired-end reads were aligned to the reference human genome (GRCh37) using Burrows–Wheeler Aligner, BWAmem (v0.7.12) and post-processed following GATK best practices 3.7. Similarly to the CoMMpass work flow, somatic SNVs were called with three different variant callers: MuTect2[53], CaVEMan[54], and MuSE[55]. Small insertions–deletions was called with MuTect2. To create the confident variant list, we chose the variants called by at least two algorithms.

**Single-cell RNA sequencing.** A recently published cohort of single-cell RNA sequencing from 29 plasma cell dyscrasia patients and 11 controls was interrogated for AID expression in single cells (GSE117156)[36].

**VDJ and HCDR3 reconstruction from WGS data.** VDJ gene segment usage and HCDR3 sequences were reconstructed from short reads data using custom scripts (Supplementary Methods).

**Molecular time.** The relative timing of each multi-gain event was estimated using the recently published R package mol_time (https://github.com/nicos-angelopoulos/mol_time)[15]. Briefly, we selected all chromosomal duplications with >50 clonal mutations estimated by the Dirichlet Process (DP). These mutations were divided using a mixed model (mclust R function) in duplicated (i.e., present on more than one allele) and non-duplicated (i.e., present on one single allele). Next, we used the corrected ratio between duplicated and non-duplicated to estimate the relative time of each large chromosomal gain acquisition (i.e., molecular time). The relative confidence interval for each molecular time was estimated using a bootstrapping function. To define the time windows in which different gains were acquired, we performed a multiple hierarchical clustering for each bootstrap solution (hclust R function) and integrated the most likely results with the Battenberg CNA changes over the time.

The presence of two duplicated copies of the same allele (tetraploidy) can be explained by two gains occurring in close temporal succession or in two distinct time windows. These situations can be differentiated based on the presence or absence of a distinct intermediary stage, with mutations present on two out of three alleles (intermediate variant allele fraction 50%). The presence of an intermediary stage indicates that sufficient time has passed between the two gains to acquire a distinguishable cluster of mutations. Conversely, the absence of an intermediary stage indicates that both gains occurred within the same time window.

**Mutational signature analysis.** The mutational signature analysis was performed for SBS following three steps as recently described[18]: (1) mutational signature de novo extraction; (2) assignment; and (3) fitting. In the first step, we ran both the hierarchical Dirichlet process (hdp: https://github.com/nicolaroberts/hdp) and SigProfiler[9] workflow to extract the involved known and potentially unknown SBS mutational signatures. hdp was run with four independent posterior sampling chains followed by 20,000 burn-in iterations, and the collection of 200 posterior samples off each chain with 200 iterations between each. SigProfiler was run with 1000 total iteration. To further validate the definition of which mtational signatures

are involved in MM and the robustness of the extracted mutational process, we analyzed the five-nucleotide context using both approaches.

The second step consisted in the assignment of each extracted process to one specific mutational signature for SBS included in the recently updated COSMIC catalogs (https://cancer.sanger.ac.uk/cosmic/signatures/SBS/)[9]. The mutational signature assignment approach was comprised of two steps[18]. In the first, all mutational signatures extracted by hdp and SigProfiler were assigned to one or a combination of two COSMIC signatures[9]. To do so, cosine similarities between the extracted mutational signatures and each COSMIC signature, or a linear combination of two COSMIC signatures (using non-negative least squares R package NNLS), were computed. In the second, we determined whether the exclusion of each extracted mutational signature would affect the reconstruction error, i.e., the difference between the original catalogs and the fitted linear combination of mutational signatures for each sample.

In the final step, we created a fitting algorithm that fits the entire mutational catalog of each patient with the mutational signatures selected through the first two steps. In this process, we reconstructed the 96-mutational profile for each sample excluding one mutational signature after another. The least contributing mutational signature was sequentially censored for that sample if removal reduced the cosine similarity <0.01. As a reference for SBS-MM1, we used the 96-class profile from hdp de novo extraction. Although the profiles extracted by SigProfiler and hdp were highly similar, the profile from hdp had a lower background contribution. Thus using the hdp-defined profile increases the specificity of SBS-MM1 calling by avoiding bleeding from other mutational signatures. SBS1 and SBS5 were always included, considering that they have been reported to be active in all normal and tumor tissue types. In this way, we reduced the bleeding of mutational signatures between samples and quantified the contribution of each mutational process using the most recent reference[9]. Importantly, because each sample is analyzed individually, the results will be reproducible irrespective of other samples included in the cohort. CIs were generated by drawing 1000 mutational profiles from the multinomial distribution, each time repeating the mutational signature fitting procedure, and finally taking the 2.5th and 97.5th percentile for each mutational signature. Mutational signature transcriptional strand bias analysis was performed by applying the Poisson test function in R, incorporated into mmsig. The source code of mmsig is available on GitHub: https://github.com/evenrus/mmsig.

All three steps for mutational signature analysis (de novo extraction, assignment, and fitting) were applied to recently published data from human pluripotent stem cell lines exposed to melphalan and to 15 controls[22].

To describe the timeline of each mutational signature, we ran mmsig on different groups of mutations acquired within different time windows. First, we analyzed clonal vs. subclonal mutations defined by DP. Thereafter, we focused on the clonal mutations only; separating them in duplicated vs. non-duplicated using the molecular time function. For the first analysis only, patients with more than one sample were considered. Overall, only clusters with >50 mutations were considered.

We interrogated the mutational signature landscape and the SBS5 contribution of 89 CLL and 107 B cell lymphoma genomes included in the ICGC/PCAWG consortium (EGAS00001001692)[56]. To quantify the contribution of each involved mutational signature, we ran mmsig including only mutational processes active in these cancers[9]. In lymphomas, the correlation between age and SBS5 was completely lost among patients with >10,000 mutations[8].

To divide CoMMpass data in high and low APOBEC, we used the 4th quartile of APOBEC mutational burden among patients with evidence of APOBEC activity (49 mutations)[26].

**Estimating patient age when landmark events occurred.** To demonstrate the existence of a clock-like mutational process in MM, we explored the association between SBS5/SBS1 and patient's age by a linear regression model (lm R function) across a large cohort of newly patients with available whole-exome sequencing data (CoMMpass; NCT01454297) and across WGS data from B cell lymphoproliferative disorders (i.e., CLL, MM, and B cell non-Hodgkin lymphoma). Next, following a recently published statistical workflow[4,5,38,39], we used an LME model (lmer R function) to estimate the mutation rate per year using CoMMpass samples collected at different time points for 72 patients. The same statistical approach was used on our WGS cohort. Here we considered each evolutionary trajectory independently based on phylogenetic tree reconstruction. This is important to avoid overestimating the mutation rate by pooling together clones that have accumulated mutations in parallel. Next, we estimated the mutation rate per year for each patient using LME models to time the MRCA and different multi-gain time windows. Two tumors in our patients had >10,000 SNVs (MRC006BBM and V0D57H) and were excluded from this analysis. An additional sample (PD26414) was excluded considering its ploidy >4 due to two independent whole-genome duplications[15]. For the MRCA, we used the fraction of clonal and shared SBS5 variant for each patient with multiple samples collected at different time points (n = 25). For the timing of different multi-gain events, we calculated the SBS-based molecular time of the first (and second) multi-gain events. To transform the relative timing of multi-gain events to absolute time, we repeated the molecular time workflow using only SBS5 mutational burden pre- and post-gain. To improve the accuracy of timing estimates, we collapsed together all trisomies acquired in the

same time window into a single multi-gain event and included only such events involving >100 SBS5 mutations.

To test the sensitivity of our approach to the assumption that SBS5 is constant over time, we included a quadratic term for age, effectively allowing for the mutation rate to parabolically increase with age.

To investigate the impact of uncertainty in mutational signature deconvolution, we included in the *lmer* and time estimation both the 2.5% and 97.5% CI of each SBS5 estimate generated by *mmsig*.

The full analytical process written in R is provided in Supplementary Data 6, 8, and 9.

**The topography of mutational signatures in MM.** Mutational catalogs from all patients with WGS were combined and divided into 1-Mb bins across the genome. For each bin, mmsig was applied for mutational signature fitting as described above. Pearson correlation was applied to assess the pairwise relationships between the mutation counts for each mutational signature and normalized genomic features across all bins. Chromatin accessibility (GM12878), replication time (GM06990), and fragile sites data were obtained from ENCODE; ALU elements from repeatmasker; and primary myeloma H3K27ac data as previously published[57].

**Data analysis and statistics**. Data analysis was carried out in R version 3.6.1. Standard statistical tests are mentioned consecutively in the manuscript while more complex analyses are described above. All reported *p* values are two sided, with a significance threshold of <0.05.

## Data availability

Sequence files are available at the European Genome-phenome archive under the accession codes: EGAD00001003309 WGS data from 30 multiple myeloma patients; phs000348.v2.p1 WGS data from 22 multiple myeloma patients; EGAS00001001692: WGS data from 89 chronic lymphocytic leukemia and 107 B cell lymphomas; phs000748.v1.p1: WXS and bulk RNA sequencing data from 933 and 792 multiple myeloma patients, respectively (CoMMpass trial); EGAS00001003709 WXS data from 40 patients with relapsed/refractory multiple myeloma; GSE117156: single-cell RNA sequencing data from 29 newly diagnosed patients and 11 control donors.

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

## Acknowledgements

This work is supported by the Memorial Sloan Kettering Cancer Center NCI Core Grant (P30 CA008748), the Instituto de Salud Carlos III (project PMP15/00007, to F.N., E.C.), the "la Caixa" Foundation Grant No. HR17-00221 (Health Research 2017 Program, to F.N., E.C.), the Ministerio de Economía y Competitividad (MINECO) SAF2013-45836-R (to E.C.) from Plan Nacional de I+D+I, Generalitat de Catalunya Suport Grups de Recerca AGAUR 2017-SGR-1142 (to E.C.), the European Regional Development Fund "Una manera de hacer Europa," Department of Veterans Affairs Merit Review Award I01BX001584-01 (to N.M.), and NIH grants P01-155258 (to H.A.-L., P.J.C., K.C.A., N.M.) and 5P50CA100707-13 (to N.M., H.A.-L., K.C.A.). F.M. is supported by the International Myeloma Society (IMS), American Society of Hematology, the International Myeloma Foundation, and The Society of Memorial Sloan Kettering Cancer Center. N.B. is funded by the University of Milan (project 22597 - PSR2017_DIP_032) and by the European Research Council under the European Union's Horizon 2020 research and innovation programme (grant agreement no. 817997). X.S.P. is supported by the Ministerio de Economía y Competitividad Grant No. SAF2017-87811-R. F.N. is supported by a pre-doctoral fellowship of the MINECO (BES-2016-076372). E.C. is supported by ICREA under the ICREA Academia programme. The work of R.S. in the framework of the ICGC MMML-Seq and derived mutational signatures has been funded by BMBF (FKZ 01KU1002A, 01KU1505 and 031A428H).

## Author contributions

F.M. designed the study, collected and analyzed data, and wrote the paper; E.R. and V.Y. collected and analyzed data and wrote the paper; D.L. and P.J.C. collected and analyzed data; N.B. analyzed data and wrote the paper; F.N., N.A., K.J.D., T.J.M., R.O., B.Z., and G.L. analyzed data; C.C., V.M., P.C., P.M., K.C.A., E.P., L.A., X.S.P., E.C., R.S., H.A.L., O.L., and N.M. collected data.

## Competing interests

The authors declare no competing interests.
