## [Peer Review File · Nature Communications]

Reviewers' comments:

Reviewer #1 (Remarks to the Author): Expertise in clonal evolution / bioinformatics

Overview

The paper by Maura and colleagues presents a genomic analysis of whole genome sequencing (WGS) and whole exome sequencing (WES) from multiple-myeloma (MM) samples. The main findings appear to be (i) the possible existence of a new mutational signature associated with melphalan (used in the treatment of MM), mostly in Figure 1. (ii) that they find diverse patterns of AID/APOBEC activity across the evolutionary stages of MM development (mostly Figure 5) and (iii) that chromosomal gains occur very early in disease evolution and up to ~40 years before diagnosis.

The paper is well written, of potential interest and the data set is undoubtedly rich. However at present I believe the analysis falls short in a number of ways, outlined below, leading me to believe that in its current form it is not appropriate for publication in Nature Communications.

Main concerns:

SBS-MM1 signature.

I am not convinced that the data supports that SBS-MM1 is necessarily a "distinct mutational process", as claimed. The lack of genomes derived from single-cells and the observation that the SBS-MM1 are typically subclonal to me suggests an equally likely explanation: these could be pre-existing mutations that are differentially selected upon exposure to melphalan. The authors touch on this possibility in the discussion but appear not to have attempted any analyses to challenge their assumption that it is a distinct mutational process. I think this element of the analysis would need to be substantially improved in order to shed more evidence on the likelihood of it being a new mutational process or simply strong selection for certain subclones that were not detected at diagnosis. The fact that not all patients exposed to melphalan demonstrate the SBS-MM1 signature at least suggests some alternative explanation might be needed too. The validation of the effect from melphalan exposed single-cell expansions (which could potentially be convincing) was performed by forcing the SBS-MM1 signature on the data rather than allowing it to emerge naturally. No measure of "confidence" is ever presented in the amplitude of the putative signature. This part of the analysis - which I think is crucial for their conclusions - was also apparently lacking from the methods / supplementary information. For example Supplementary table 5 did not appear to be included and Supplementary Figure 1E does not show the relative amplitudes of the two signatures (there were multiple instances of this). From this is really is not possible to judge how robust this analysis was and at present I do not think the authors have convincingly shown it must be a distinct mutational process.

Early chromosomal gains

The analysis on the timing of chromosomal gains is interesting but the conclusions do not seem that novel. I believe the authors published a paper in Nature Communications earlier this year where the same feature was pointed out (i.e. chromosomal gains are some of the earliest events). Admittedly they have here put some chronological time on these estimates using the SBS5 signature as a clock, and come up with numbers in the region of 30-40 years but I found the analysis of the SBS5 clock itself a little incomplete. In the cases where the authors had samples from multiple time points, was the rate of the clock consistent i.e. did it appear to tick at a constant rate within an individual? Is the person-to-person variation in the rate of the SBS5 clock here consistent with other cancer studies and studies in healthy individuals? If not, how much more variation is there? The variation around the mean clock rate is substantial so would affect timing estimates by a lot and at present the authors do not challenge their numbers enough. I feel a more thorough analysis of the SBS5 clock is needed in order to determine how much to believe the timings presented in Figure 7. It would also be important to validate these claims in samples obtained from MGUS patients or similar.

Patterns of AID/APOBEC activity.

I am unconvinced of some of the claims about AID/APOBEC activity. While I can see that SBS9 (AID) activity appears to be enriched in the early phase of the disease (clonal mutations) the claims about there being 4 distinct processes governing AID/APOBEC activity (Figure 5E) are not particularly convincing and again no confidence measure on whether there really are 4 clusters of whether it is largely noise. I am also not sure the analysis of the relative contribution of APOBEC3A vs B (Figure 2) which appears to be from data across solid cancers fits in with the rest of the paper.

Reviewer #2 (Remarks to the Author): Expertise in MM

This is an interesting paper if complicated paper. The authors have analyzed a set of MM sequencing data for signatures. The analysis is original and contains much interesting information. The data set is not large but adequate for their purposes. They have accessed whole genome and whole exome data but the analysis would benefit from more whole genome data. The data set includes patients at presentation and at relapse. The data they present is credible and adds significantly to the MM literature.

- i. They develop a clock mutation to define when the MM clone is first established.
- ii. They use mutational signatures to understand the mutational processes shaping the MM genome and when they occur.
- iii. They define a signature of melphalan exposure which may be driving relapse post stem cell transplantation.
- iv. They identify a non canonical aid signature which they use to imply a mechanistic basis for how MM develops in relation to the germinal center.

These features are important to our understanding of MM pathogenesis.

A negative feature is that they include so much information that they do not focus on any one of the stories and develop it fully. Despite this I can understand why they have done this because all of these approaches rely upon the use of mutational signatures. Overall this is an excellent paper that makes a significant contribution and will be extensively referenced.

Reviewers' comments:

Reviewer #1 (Remarks to the Author): Expertise in clonal evolution / bioinformatics

Overview

The paper by Maura and colleagues presents a genomic analysis of whole genome sequencing (WGS) and whole exome sequencing (WES) from multiple-myeloma (MM) samples. The main findings appear to be (i) the possible existence of a new mutational signature associated with melphalan (used in the treatment of MM), mostly in Figure 1. (ii) that they find diverse patterns of AID/APOBEC activity across the evolutionary stages of MM development (mostly Figure 5) and (iii) that chromosomal gains occur very early in disease evolution and up to ~40 years before diagnosis.

The paper is well written, of potential interest and the data set is undoubtedly rich. However at present I believe the analysis falls short in a number of ways, outlined below, leading me to believe that in its current form it is not appropriate for publication in Nature Communications.

Main concerns:

SBS-MM1 signature.

I am not convinced that the data supports that SBS-MM1 is necessarily a “distinct mutational process”, as claimed. The lack of genomes derived from single-cells and the observation that the SBS-MM1 are typically subclonal to me suggests an equally likely explanation: these could be pre-existing mutations that are differentially selected upon exposure to melphalan.

The authors touch on this possibility in the discussion but appear not to have attempted any analyses to challenge their assumption that it is a distinct mutational process. I think this element of the analysis would need to be substantially improved in order to shed more evidence on the likelihood of it being a new mutational process or simply strong selection for certain subclones that were not detected at diagnosis. The fact that not all patients exposed to melphalan demonstrate the SBS-MM1 signature at least suggests some alternative explanation might be needed too.

The validation of the effect from melphalan exposed single-cell expansions (which could potentially be convincing) was performed by forcing the SBS-MM1 signature on the data rather than allowing it to emerge naturally. No measure of “confidence” is ever presented in the amplitude of the putative signature. This part of the analysis - which I think is crucial for their conclusions - was also apparently lacking from the methods / supplementary information. For example, Supplementary table 5 did not appear to be included and Supplementary Figure 1E does not show the relative amplitudes of the two signatures (there were multiple instances of this). From this is really is not possible to judge how robust this analysis was and at present I do not think the authors have convincingly shown it must be a distinct mutational process.

The reviewer raises two main concerns regarding the mutational signature SBS-MM1: 1) is SBS-MM1 a distinct mutational signature; and 2) if yes, is there sufficient evidence to establish a causal relationship between SBS-MM1 mutations and melphalan.

Addressing the first question, our mutational signature discovery analysis followed current best-practices, applying two independent mutational signature *de novo* extraction algorithms: SigProfiler and the hierarchical Dirichlet process (hdp). SigProfiler has been recently tested on a large cohort (>10,000) of solid and hematological cancers with available WGS and WES data

(PCAWG study, Alexandrov et al Biorxiv 2018). No multiple myeloma WGS data was included in this study and SBS-MM1 was not extracted in any sample or tumor type. In contrast, SigProfiler was able to extract SBS-MM1 in our multiple myeloma WGS cohort considering both the 96 and 1536 classes. The existence of this mutational process was further validated running the hdp. Overall, we believe that these data support the existence of SBS-MM1 as a new mutational process in multiple myeloma.

Turning to the question of causality, two lines of evidence support that SBS-MM1 is a chemotherapy-induced mutational signature specifically caused by melphalan: 1) mutational signature analysis of patient samples, and 2) independent confirmation of SBS-MM1 in human-induced pluripotent stem cells (iPSCs) exposed to melphalan.

First, across three independent patient cohorts (one with WGS and two with WES), SBS-MM1 could only be identified in samples obtained after melphalan treatment. Importantly, SBS-MM1 was not only absent in newly diagnosed and smoldering MM, but also absent in relapsed patients who had not received melphalan as part of their treatment. This association was particularly striking looking at the SBS-MM1 activity over time after reconstructing the phylogenetic tree of each patient (**Figure 4A** and **4C**). This mutational process was observed only among clusters of late clonal/subclonal mutations selected after treatment with melphalan. Interestingly all but one of these clusters showed a strong transcriptional strand bias, a feature usually associated with chemotherapy related signatures (e.g. platinum-SBS31 and 35). One patient in our study (PD26403) represents an emblematic example of the potential causal relationship between melphalan and SBS-MM1. This is the only patient with samples collected during the pre-malignant phase (smoldering multiple myeloma), at diagnosis and at relapse after high dose melphalan and transplant. Branching evolution was observed in both progressions (i.e. from smoldering to myeloma and from myeloma diagnosis to relapse). However, SBS-MM1 was identified only at relapse and not in the previous evolution and selection. Similar data were observed in the CoMMpass cohort and are reported in **Figure 4D**. Here, thanks to the detailed clinical annotation of the CoMMpass cohort, we reconstructed the phylogenetic tree of all patients with samples collected at baseline and at first relapse, dividing these patients in melphalan-exposed and non-exposed. SBS-MM1 was observed only among mutations positively selected after transplant (i.e mutations not detectable before treatment and selected at relapse), but was not detected among clonal mutations (i.e. mutations clonal in both samples), mutations positively selected after treatment without high dose melphalan, or negatively selected mutations. This suggests that SBS-MM1 is not associated with treatment or clonal sweeps in general, but specifically associated with melphalan.

It is well-known that the application of specific therapies can lead to selection of resistance-conferring mutations in key genes. In multiple myeloma, however, there are no known mutations that specifically cause resistance to melphalan. Furthermore, we did not find any association between SBS-MM1 and specific genomic alterations, in contrast to the association between *IGH-MAF/MAFB* translocations and APOBEC activity. We are not aware of previous reports suggesting that mutations are selected by a specific therapy based solely on their trinucleotide context. Conversely, the concept of chemotherapy-induced mutational signatures has been established in vitro and in vivo in two recent major works: Kucab et al Cell 2019 and Pinch et al Nat Gen 2019.

These concepts and data are now included in the new version of the manuscript (page 7, line 143-145; page 7, line 152-154 and page 12, line 267-276) and in the new **Supplementary Figure 9**. The absence of correlation between melphalan and the most frequent clinical and genomic features is reported in the new **Supplementary Table 4**.

As pointed out by the reviewer, not all patients exposed to melphalan have detectable SBS-MM1. We agree that the previous discussion and **Figure 8a** did not provide a clear and comprehensive interpretation of this observation. We believe there are two main reasons why SBS-MM1 may be undetectable despite prior melphalan exposure. The first is based on the “single-cell expansion” model, recently described in an elegant study from the Lopez-Bigas group (Pinch et al Nat Gen 2019). In this study, authors showed how platinum and other chemotherapeutic agents are able to induce private mutations in each surviving cell after the first round of replication, but a single cell must expand in order for these mutations to become detectable. The second potential escape mechanism is specific to hematological cancers treated with autologous stem cell transplant and has not been described in previous studies focusing on solid tumors. It is known that the stem cell product collected from multiple myeloma patients can be contaminated by tumor cells (Wuillème S et al. Bone Marrow Transplant 2016). These cells are collected before melphalan treatment and re-infused after. This provides a potential explanation for the absence of SBS-MM1, if relapse arises from tumor cells in the graft. **In the new version of the manuscript we provide a better description of this concept (page 19, line 460-468) and we include the myeloma post-transplant engraftment model in the new Figure 8A (top).**

The second line of evidence supporting a causal relationship between SBS-MM1 and melphalan was derived from re-analyzing data from a recent publication by the Nik-Zainal group (Kucab et al Cell 2019), where human pluripotent stem cells were exposed to a series of mutagenic agents, including melphalan. We thank the reviewer for pointing out methodological weaknesses in this part of our analysis, including the reporting of results in **Supplementary Figure 1E**. We agree that a *de novo* extraction should always be performed. Therefore, in the new version of the manuscript, the melphalan exposed cell line and 15 controls from Kucab et al (Cell 2019) were investigated by both hdp and SigProfiler together with all MM and CLL WGSs. The CLL cohort was included to correct and estimate the inter-bleeding of signatures in these *de novo* extractions. In fact, CLL is never treated with high-dose melphalan and the cohort included in this study has been extensively investigated by many different algorithms and groups over the years, without identifying any mutational signatures that resemble SBS-MM1. Interestingly, both algorithms identify SBS-MM1 only in the melphalan exposed single cell culture and in the previously described 9 MM patients. No significant SBS-MM1 contribution was detected among CLLs and control cell lines. **In the new version of the manuscript these new data are discussed (page 7, line 158-164) and reported in the new Supplementary Figure 2, including 95 % confidence intervals for the estimated mutational signature contributions.**

Following our proposed workflow, we also applied the “assignment” and “fitting” of signatures on these 16 cell lines (1 melphalan exposed and 15 controls; Kucab et al; Cell 2019). We tested all possible paired combinations of signatures, including SBS-MM1 and the “Control signature” (i.e. the signature detected in all single cell expansions exposed to different agents; Kucab et al, Cell 2019). The combination of SBS-MM1 and “Control” showed the highest cosine similarity compared to all other combinations. In contrast, the combination of SBS-MM1 and Control signature did not significantly explain any of the 15 control cell lines. Finally, mmsig (<https://github.com/evenrus/mmsig>) identified SBS-MM1 only in the melphalan exposed cell line, but not in the controls. Taken together, extensive re-analysis of the *in vitro* data from Kucab et al supports our conclusion that SBS-MM1 is a melphalan-induced mutational signature. **This comprehensive analysis is now fully reported in the new version of the manuscript (pages 7-8, line 164-169), in the new Supplementary Table 7 and in the new Supplementary Figure 3 and 4. The methodological workflow on the single-cell exposed cell lines is now described in the**

new paper methods. The full R code used for the assignment of signatures is now included as **Supplementary Data S1**.

Regarding our mutational signature analytical workflow and the possible uncertainty, in contrast to other fitting algorithms, *mmsig* (package: <https://github.com/evenrus/mmsig>) provides also the 95 % confidence intervals for each signature contribution based on re-sampling from the multinomial distribution of each mutational signature profile. In the new version of the manuscript, 95% confidence intervals of all estimations for each patient are now reported in the new **Supplementary Table 3**. The SBS-MM1 transcriptional strand bias for each case is fully reported in the new **Supplementary Table 5** and **6**. The main mutational data of each patient are also included in the github depository as part of the tutorial.

Early chromosomal gains

The analysis on the timing of chromosomal gains is interesting but the conclusions do not seem that novel. I believe the authors published a paper in Nature Communications earlier this year where the same feature was pointed out (i.e. chromosomal gains are some of the earliest events). Admittedly they have here put some chronological time on these estimates using the SBS5 signature as a clock, and come up with numbers in the region of 30-40 years but I found the analysis of the SBS5 clock itself a little incomplete. In the cases where the authors had samples from multiple time points, was the rate of the clock consistent i.e. did it appear to tick at a constant rate within an individual? Is the person-to-person variation in the rate of the SBS5 clock here consistent with other cancer studies and studies in healthy individuals? If not, how much more variation is there? The variation around the mean clock rate is substantial so would affect timing estimates by a lot and at present the authors do not challenge their numbers enough. I feel a more thorough analysis of the SBS5 clock is needed in order to determine how much to believe the timings presented in Figure 7. It would also be important to validate these claims in samples obtained from MGUS patients or similar.

We agree with the reviewer that the molecular time analysis is an extension of our recently published work. However, the aims of the two analyses are different. In our first paper we performed a relative molecular timing of large gains to show how the hyperdiploid cytogenetic profile of multiple myeloma reflects the sum of multi-gain events acquired in different time windows. Here, we used the same concept to perform an absolute time estimation based on the SBS5 clock like mutational process.

The concept of clock-like mutational processes was first introduced by Alexandrov et al (Nat Gen 2015) and has since then been applied to several cancers and normal tissues. In this study we applied an updated version of the statistical workflow recently published for kidney cancer (Mitchell T. Cell 2018), lung cancer (June-Koo Lee et al. Cell 2019), normal endometrium (Moore L Biorxiv 2018 - <https://doi.org/10.1101/505685>) and normal colorectal cells (Lee HS, Nature 2019). In all these studies a linear mixed effect model was used on patients with and without multiple samples to estimate the global and patient-specific mutation rate. Linear mixed effect models draw on multiple observations from the same individual to increase the accuracy of mutation rate estimates.

Since the first observations (i.e. Alexandrov et al Nat Gen 2015 and Blokzijl et al Nature 2015), the mutation rate was shown to differ from one tissue or disease to another, as well as between individuals. It is not straightforward to directly compare the results from one study with another, but a reasonable comparison is the study by Mitchell et al. (Cell 2019). They performed a similar analysis to our study with a similar cohort of WGSs (multiple samples from same patients)

sequenced in the same institution and with a similar coverage. Mitchell et al used the global mutation rate as a clock, as opposed to SBS5 alone in our study, and they reported a higher clock-rate in absolute numbers (89 vs. 39 mutations per year). However, the magnitude of between-patient variation was similar, with a between-patient standard deviation 17 mutations/year (of 19 % of the cohort average), as compared with 7 mutations/year in our study (18 % of the cohort average).

Re-analyzing their data (see below; Peter J Campbell, personal communication) reveals a between-patient variation that is highly similar to our study (reported in the new **Supplementary Figure 11** in this paper).

Overall, each patient has their own evolutionary history, and, similarly to other tumor and normal cells, different mutation burdens between patients are expected. However, the overall trend shown by the linear mixed effects models support a constant SBS5 mutation rate, similarly to many other cancer types and normal cells.

In the new version of the manuscript we further analyze the impact of the main potential confounding factors on the SBS5 mutation rate, including:

- Sample coverage
- Sample Purity
- Disease stage
- Ploidy
- Uncertainty in mutational signature estimation (i.e. *mmsig* 95% confidence intervals)
- Potential impact of SBS-MM1

Despite attempts at improving our linear mixed effects model for mutation rate by including additional factors, none of the more complex models performed significantly better than the simplest model considering only SBS5 mutation burden and age. Moreover, the different models provided very similar estimates of SBS5 mutation rate. **Comprehensive exploratory analysis and**

model comparisons are included together with the whole computational analysis for our clock analysis in the new **Supplementary Data S3**, in the new version of the manuscript (pages 16-17 line 379-408) and in the new **Supplementary Figure 11-14**.

We agree with the reviewer that a longitudinal investigation between two samples collected at different time points from the same individual could further elucidate the clock-like properties of different mutational processes. However, such an analysis would require a long time-interval between the samples, and the median time between sample collections in our study was 1.5 years. Considering a median mutation rate of 39 SBS5 mutation per year, this interval is too short for a proper longitudinal investigation for each patient. Furthermore, as explained in the methods and in the new **Supplementary Data S3**, similarly to Mitchell et al. Cell 2019, we did not use the individual sample mutational burden, but we considered each evolutionary trajectory independently based on phylogenetic tree reconstruction. This is important to avoid over-estimating the mutation rate by pooling together clones that have accumulated mutations in parallel. Thus, the main benefit from including more than one sample from the same patient in this study was not specifically to allow for mutations to accumulate over time, but to increase the accuracy of phylogenetic tree re-construction.

In the new version of the manuscript we report all the SBS5 mutational rate estimates per year to show the relatively low difference for most of the patients (**Supplementary Figure 11**). The entire molecular clock analytical workflow is also reported and explained in the new **Supplementary Data 2** and **Supplementary Data 3**.

We agree with the reviewer that the inclusion of MM precursors would be important, and in fact our cohort included 10 patients with paired samples collected at the time of smoldering multiple myeloma diagnosis and at the time of progression to symptomatic MM. The estimated SBS5 mutation rate in these patients did not differ from the other patients in our study. **These data are included in the new Supplementary Table 1, Supplementary Figure 11 and Supplementary Data S3.**

Patterns of AID/APOBEC activity.

I am unconvinced of some of the claims about AID/APOBEC activity. While I can see that SBS9 (AID) activity appears to be enriched in the early phase of the disease (clonal mutations) the claims about there being 4 distinct processes governing AID/APOBEC activity (Figure 5E) are not particularly convincing and again no confidence measure on whether there really are 4 clusters of whether it is largely noise. I am also not sure the analysis of the relative contribution of APOBEC3A vs B (Figure 2) which appears to be from data across solid cancers fits in with the rest of the paper.

We agree with the reviewer that the previous **Figure 5E** (heatmap) reported a descriptive analysis not supported by a robust statistical workflow. The notion that APOBEC and AID are strongly involved in different phases of MM development has been supported considering two independent cohorts of patients and it has been suggested in other previous studies. Despite our limited WGS cohort, we provide several lines of evidence regarding the existence of at least 4 distinct mutational signatures' evolutionary patterns. **In the new version of the manuscript we better explain and describe these data (page 15; line 342-346. The Figure 5E heatmap has been replaced by a cartoon summarizing the 4 main patterns observed in our mutational signature temporal investigation.**

Regarding the APOBEC3A/3B investigation we believe that this is an important aspect to mention in the paper because it highlights the unique APOBEC activity in patients with *MAF/MAFB* translocations and its similarity to the one observed in hyper-APOBEC solid

cancers. This consideration is now included in the new version of the manuscript (page 9, line 211-212)

Reviewer #2 (Remarks to the Author): Expertise in MM

This is an interesting paper if complicated paper. The authors have analyzed a set of MM sequencing data for signatures. The analysis is original and contains much interesting information. The data set is not large but adequate for their purposes. They have accessed whole genome and whole exome data but the analysis would benefit from more whole genome data. The data set includes patients at presentation and at relapse. The data they present is credible and adds significantly to the MM literature.

- i. They develop a clock mutation to define when the MM clone is first established.
- ii. They use mutational signatures to understand the mutational processes shaping the MM genome and when they occur.
- iii. They define a signature of melphalan exposure which may be driving relapse post stem cell transplantation.
- iv. They identify a non canonical aid signature which they use to imply a mechanistic basis for how MM develops in relation to the germinal center.

These features are important to our understanding of MM pathogenesis.

A negative feature is that they include so much information that they do not focus on any one of the stories and develop it fully. Despite this I can understand why they have done this because all of these approaches rely upon the use of mutational signatures. Overall this is an excellent paper that makes a significant contribution and will be extensively referenced.

We thank the reviewer for their considerations. In the new version of the manuscript we provided additional explanations and material to improve the data flow and to highlight their significance.

REVIEWERS' COMMENTS:

Reviewer #3 (Remarks to the Author):

This manuscript presents an important novel contribution to our genomic view of MM evolution. To me, the manuscript is long, but well written, and the results and conclusions point toward a novel signature that could help improve the estimation of important properties of the disease. The mechanisms implied in the process of disease development are still largely unclear to me after reading this paper, but the argument made to use molecular data to support a first idea of the tumor-age at diagnosis is intriguing. I could not find any major flaws in their analysis. While most figures are complicated to interpret on first glance, which speaks to the technical nature of this report, the final main figure resolved some of this obstacle, as it provided some closure of the story this paper attempts to tell. I think this dense work deserves to be published.

REVIEWERS' COMMENTS:

Reviewer #3 (Remarks to the Author):

This manuscript present an important novel contribution to our genomic view of MM evolution. To me, the manuscript is long, but well written, and the results and conclusions point toward a novel signature that could help improve to estimate important properties of the disease. The mechanisms implied in the process of disease development is still largely unclear to me after reading this paper, but the argument made to use molecular data to support a first idea of the tumor-age at diagnosis is intriguing. I could not find any major flaws in their analysis. While most figures are complicated to interpret on first glance, which speaks to the technical nature of this report, the final main figure resolved some of this obstacle, as it provided some closure of the story this papers attempts to tell. I think this dense work deserves to be published.

We thank the reviewer for his/her considerations.